# Learning Semantic Representations to Verify Hardware Designs

**Shobha Vasudevan** *
Google Research, Brain Team
shovasu@google.com

**Wenjie Jiang** *
Google Research, Brain Team
wenjiej@google.com

**David Bieber**
Google Research, Brain Team
dbieber@google.com

**Rishabh Singh**
Google X
rising@google.com

**Hamid Shojaei**
Google
hamids@google.com

**Richard Ho**
Google
riho@google.com

**Charles Sutton**
Google Research, Brain Team
charlessutton@google.com

## Abstract

Verification is a serious bottleneck in the industrial hardware design cycle, routinely requiring person-years of effort. Practical verification relies on a "best effort" process that simulates the design on test inputs. This suggests a new research question: Can this simulation data be exploited to learn a continuous representation of a hardware design that allows us to predict its functionality? As a first approach to this new problem, we introduce Design2Vec, a deep architecture that learns semantic abstractions of hardware designs. The key idea is to work at a higher level of abstraction than the gate or the bit level, namely the Register Transfer Level (RTL), which is similar to software source code, and can be represented by a graph that incorporates control and data flow. This allows us to learn representations of RTL syntax and semantics using a graph neural network. We apply these representations to several tasks within verification, including predicting what cover points of the design will be covered (simulated) by a test, and generating new tests to cover desired cover points. We evaluate Design2Vec on three real-world hardware designs, including the TPU, Google's industrial chip used in commercial data centers. Our results demonstrate that Design2Vec dramatically outperforms baseline approaches that do not incorporate the RTL semantics and scales to industrial designs. It generates tests that cover design points that are considered hard to cover with manually written tests by design verification experts in a fraction of the time.

## 1 Introduction

Hardware designs are verified to check if a design implements the architectural specification correctly. Verification is widely considered the most serious bottleneck in the contemporary industrial hardware design cycle [2, 16, 15]. It requires 60-75% of the time, compute, and human resources during the design phase, routinely taking multiple person-years of effort. Verification is a complex problem because modern hardware has billions of inputs and flops, and the number of states is exponential in the number of flops and inputs. Checking every state of the system is infeasible due to a combinatorial

---

*Equal contributors

35th Conference on Neural Information Processing Systems (NeurIPS 2021).

state space explosion. While there is excellent research in automatic hardware verification techniques involving formal and static analysis, their applicability in practice is limited by scale. Instead, practical design verification sacrifices automation and completeness for a "best effort, risk reducing" process based on simulating the design on test inputs.[2]

Traditionally, formal verification methods analyze synchronous hardware designs through reachability analysis of a gate level state transition graph. Each node in this graph corresponds to a single value of bit-level state of all registers, and edges correspond to the legal changes in state that the design can make in a single clock cycle. It is well known in the formal methods community that most questions about hardware functionality can be posed as reachability questions in the state transition graph. This suggests a fundamental new research question in representation learning: *Can we learn a continuous representation of a hardware design that allows us to predict its functionality?*

This paper presents a first approach to this problem. Since designs are practically verified by simulations using millions of test inputs, this gives us a ready source of training data. To avoid the combinatorial explosion at the gate level, we approach the problem at a higher level of abstraction, the Register Transfer Level (RTL) that describes hardware at the bit-vector or integer level. RTL is described in a Hardware Description Language (Verilog RTL [3]) that is syntactically similar to the source code of software, while modeling the concurrent, non-deterministic, non-terminating, and reactive semantics of hardware. Despite the higher level of abstraction, RTL static analysis approaches for reachability analysis [4] and test input generation [31] do not scale to even reasonably sized practical designs. So our research question can be restated as: *Can we use simulation data to learn a tractable continuous representation that can predict the answers to the state reachability queries in hardware?*

In this paper, we introduce Design2Vec, an architecture for learning continuous representations that capture the semantics of a hardware design. We choose to represent the design as a control data flow graph (CDFG) at the RTL level. Based on the CDFG, we use graph neural networks (GNN) to learn representations that predict states reached by the design when simulated on test inputs. While standard GNN architectures do work well, we achieve improved performance by introducing a new propagation layer that specifically incorporates the concurrent and non-terminating semantics of RTL. Design2Vec is trained to predict if, given a simulation test input, a particular branch (like a case statement in software) will be covered. The CDFG is an abstraction of the gate-level state space since one edge in the CDFG maps to many edges in the gate-level transition graph. We can therefore interpret Design2Vec as learning an *abstraction* of the intractable gate-level design state space.

We apply Design2Vec to two practical problems in hardware verification: coverage prediction and test generation. The coverage prediction model can act as a proxy simulator, and an engineer can query it to estimate coverage in seconds, instead of waiting for a night of simulations. Our test generation method uses a gradient-based search over the trained Design2Vec model to generate new tests. It is desirable to find tests for *hard to cover* points, or points human experts find difficult to cover after reaching around 80% (or more) cumulative coverage using random testing. This coverage plateau can take months to close, effectively taking multiple expert months of productivity and effort.

We demonstrate Design2Vec is able to successfully learn representations of multiple designs: (i) IBEX v1 [1], a RISC-V design (ii) IBEX v2, IBEX enhanced with security features, and (iii) Tensor Processing Unit (TPU) [23], Google's industrial scale commercial infrastructure ML accelerator chip, and performs dramatically better than black-box baselines that are uninformed by knowledge of the design (up to 50% better, and on average 20% better for real designs). Our results show that Design2Vec achieves over 90% accuracy in coverage prediction on the industrial TPU design, making it ideal to serve as a proxy simulator that can evaluate if a test can cover a given point within seconds. Our results on test generation show that Design2Vec is able to successfully find tests for hard to cover points in real designs in a fraction of the time (up to 88 fewer simulator calls for TPU and 40 fewer for IBEX) as compared to random testing and a black-box optimizer. This order of magnitude improvement can potentially lead to huge savings. Our key contributions follow.

- We introduce the problem of learning continuous representations (abstractions) of hardware semantics that can be used for various tasks in hardware verification. We present Design2Vec, a model that encodes the syntactic and semantic structure of an RTL design using a GNN based architecture.

---

[2]All future references to verification will imply simulation based design verification as the majority practice in industry, not formal verification.

- We propose the RTL IPA-GNN, which extends a recent architecture for learning to execute software [6] to model the concurrent and non-terminating semantics of RTL.
- We introduce a test generation algorithm that uses Design2Vec to generate focused tests for a subset of cover points. The tests generated by Design2Vec are able to successfully cover hard to cover points using significantly fewer simulations than state of practice human expert guided random testing and a state of the art black-box optimization tool.
- We demonstrate the scalability of the approach to large industrial scale designs. Our approach is practical in industrial settings.

## 2   Design2Vec: Representation learning of design abstractions

Verilog RTL source code is a powerful tool for writing non-terminating, continuous, reactive "programs" to design hardware. A program in RTL is structured as a modular hierarchy, with multiple concurrent blocks executing within each module. We consider the subset of behavioral RTL that can be synthesized into gate level designs [3].

Figure 1 shows an example snippet of Verilog RTL source code. A "module" corresponds to a hardware construct (e.g. a decoder or ALU). A module can instantiate other modules (e.g. a cache module may instantiate a prefetch module). Each "variable" in the RTL corresponds to either an input signal (input), an output signal (output), a register that stores values (reg) or a temporary variable (wire). The bit width of each such variable is declared. In the example, a, b, and state are two-bit registers, c, d, p, and q are two-bit input signals.

This design has three concurrent (always) blocks denoted by three colors. For synchronous hardware, an external clock signal triggers execution of each concurrent block per *clock cycle*. On each clock cycle, one statement in each always block is executed. The statements in an always block are thereby executed in sequence over multiple consecutive clock cycles. When executing a statement, the input values of variables in the current cycle come from the output values of variables from the previous cycle. There is an implicit loop from the end of each block back to its start, simulating the non-terminating nature of hardware. Within a cycle, the order in which the always blocks are executed is *non-deterministic*. In practice, it is as determined by an RTL simulator. Figure 4 in Appendix A.1 shows the execution (simulation) over a three cycle window of this example.

RTL designs are typically simulated using an RTL design simulator, which does not suffice for the design goals of our analysis. For use with machine learning, we choose to represent RTL as a control data flow graph (CDFG). The CDFG we construct for each Verilog RTL program encodes the program's simulation semantics, so that our models may make inferences about the their behavior.

The CDFG has nodes and heterogeneous edges. Nodes correspond to statements in code. Edges corresponds to either data flow or control flow. Dotted lines show data dependencies between concurrently executing blocks. The root node corresponds to the begin node of the top module. Branches denote localized node sequences. The designs that we consider contain branches, but not loops, which are atypical in synthesizable RTL.

A *test* is a set of high level parameters, each of which can be Boolean, integer or categorical. A test defines a distribution over input signals to the design. When a test is run, the testbench samples many inputs from this distribution, and simulates the design on that sequence of inputs. The output of the testbench is Boolean input vectors applied to the RTL design under test. These input vectors execute different design paths. All the branches executed along different design paths are said to be *covered*.

A test is written to achieve different types of design code coverage. Branch coverage tends to be the most important metric. Branch coverage of a test refers to branches executed by that test. Each branch is referred to as a *cover point*. A cover point over the RTL CDFG is identified as a sequence of nodes starting from the branch control node and ending at the destination of the branch. Two local branch cover points lie along the same global path if they have ancestor/descendant relationship. Figure 5 in Appendix A.1 shows the executed branches the input arrives at a certain clock cycle.

### 2.1   Architecture

Our Design2Vec architecture, shown in Figure 2, is trained on the supervised task of coverage prediction. The network takes as input a cover point $C$, represented as a sequence of CDFG notes, and a test parameter vector $I$, and outputs the probability $\mathsf{is\_hit}(C, I)$ that the test $I$ covers $C$.

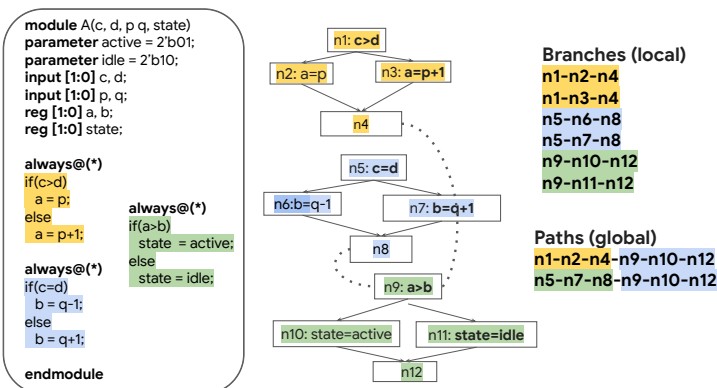

Figure 1: An example snippet of a Verilog RTL source code module and the corresponding CDFG.

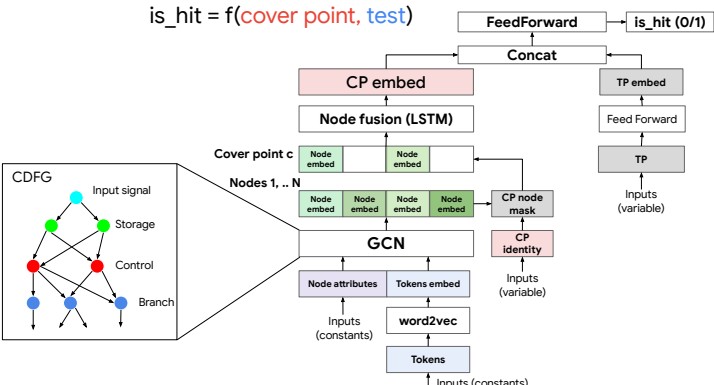

Figure 2: The Design2Vec architecture that takes as input a cover point and an input test vector, and predicts the corresponding coverage. CDFG: Control and data flow graph. TP: test parameter. CP: cover point.

**Cover point embedding:** We use a graph convolution network (GCN) over the Verilog CDFG. Each node $n \in N$ has attributes/features denoted by $f_j^n$ such as node identifier, node type, fan in and fan out. In addition, the RTL token sequence $s^n = < s_1^n, s_2^n, \cdots, s_k^n >$ in the program statement is also considered as an additional attribute. We first perform an embedding of the token sequence $\phi(s^n)$ using an LSTM (or alternatively using a pre-trained word2vec module):

$$\phi(s^n) = \mathsf{LSTM}(< s_1^n, s_2^n, \cdots, s_k^n >). \tag{1}$$

The initial embedding of a node $\phi^{(0)}(n)$ is computed as concatenation of the node attribute values and the token sequence embedding:

$$\phi^{(0)}(n) = \mathsf{Concat}(< \phi(s^n), f_1^n, \cdots, f_l^n >). \tag{2}$$

Let the embedding of a graph $G$ at a step $t$ be denoted by $\psi^{(t)}(G) = \{\phi^{(t)}(n) | \forall n \in N\}$. We use a GCN (with learnable parameters $\theta$) to perform a sequence of graph convolution steps to update the node embeddings:

$$\psi^{(t+1)}(G) = f_\theta(\psi^{(t)}(G)) \qquad \forall t \in \{0, 1, \cdots, T\}. \tag{3}$$

A cover point $C = < n_1, n_2, \cdots, n_m >$ is a sequence of nodes. The cover point identity is used to select the corresponding sequence of node embeddings of the cover point nodes. Since these sequences are of varying length, an LSTM layer is used to produce the cover point embeddings

$$\phi^{(T)}(C) = \mathsf{LSTM}(< \phi^{(T)}(n_1), \phi^{(T)}(n_2), \cdots, \phi^{(T)}(n_m) >). \tag{4}$$

**Test parameter embedding**: For the input test parameters $I$, each of which can be integers or categorical, we learn an embedding for the parameters by passing them through a feed forward MLP layer as

$$\phi(I) = \mathsf{MLP}(\mathsf{Concat}(i_1, \cdots, i_p)). \tag{5}$$

**MLP layer**: Finally, the test parameter $\phi(I)$ and cover point embeddings $\phi^{(T)}(C)$ are concatenated and provided to a feed forward layer with sigmoid activation to predict for each cover point

$$\mathsf{is\_hit}(C, I) = \mathsf{MLP}_\sigma(\mathsf{Concat}(\phi^{(T)}(C), \phi(I))). \tag{6}$$

Given a supervised dataset of test inputs and the corresponding coverage information of different cover points in the RTL design, the network is trained using binary cross entropy loss. We find that our model learns more from local paths than global paths (root to cover point). A reason for this is that GNNs are not proficient at learning long path information. In the case of a control flow, tracing and learning global paths is relevant. We use shortcut edges, where we add edges along every kth node along a path, to reinforce the relationship of long paths (results in Appendix D). We use a variation of the GNN, called the GNN-MLP [7] to propagate edge information. We also used gating layers and residual layers in the GGNN.

## 2.2 RTL IPA-GNN architecture

We enhance the recently proposed Instruction Pointer Attention Graph Neural Network (IPA-GNN) architecture [6] that explicitly models control flow in programs. This section motivates and precisely describes each of the architectural enhancements our novel RTL IPA-GNN architecture makes over the original IPA-GNN in order to model the concurrent, non-terminating semantics of RTL.

First, since RTL hardware is highly parallel, the RTL IPA-GNN maintains a separate instruction pointer for each always block in the hardware design. The original includes only a single instruction pointer. Making this change allows modeling the concurrent execution of all modules in an RTL specification. Accordingly, the RTL IPA-GNN instantiates its soft instruction pointer $p_{t,n}$ as

$$p_{0,n} = 1_{\text{n is an always block start node}},$$

and our RTL IPA-GNN implementation computes its hidden state proposals as

$$a_{t,n}^{(1)} = \text{Dense}(h_{t-1,n}).$$

Second, the domain of RTL requires the model support switch conditions, not just binary conditions. So, we modify the soft branch decision mechanism of the IPA-GNN. In the RTL IPA-GNN, the soft branch decisions at timestep $t$ are given as

$$b_{t,n,m} = \text{softmax}\left(\text{Dense}(h_{t-1,n}) \cdot \text{Embed}(e_{n,m})\right),$$

where $m \in N_{\text{out}}(x_n)$ is a control node child of $x_n$.

$\text{Embed}(e_{n,m})$ is an embedding for the control edge from $x_n$ to $x_m$. It includes an embedding of whether the condition is positive or negative, the first variable referenced by the condition, and the form of the condition. It is computed as a concatenation of a learned embedding of each of these three properties of the condition represented by the edge.

Third, data flow in an RTL design is also more complex than in a single-threaded program. We model data flow in the RTL IPA-GNN architecture by propagating messages between nodes along data flow edges at each step of the model. This entails aggregating hidden state proposals both from control node state proposals (these are within a node's always block) and from the proposals of other parent nodes (which may be from a different always block), giving

$$h'_{t,n} = \sum_{n' \in N_{\text{in}}(n) \cap N_{\text{ctrl}}} p_{t-1,n'} \cdot b_{t,n',n} \cdot a_{t,n'}^{(1)} + \sum_{n' \in N_{\text{in}}(n) \cap \bar{N}_{\text{ctrl}}} a_{t,n'}^{(1)}.$$

Here $N_{\text{ctrl}} \subseteq N$ denotes the set of control nodes in the CDFG.

Finally, we introduce a control edge from the end of each always block to the start of the always block to model non-termination, a critical RTL property. With this update to the CDFG, the instruction pointer probability mass in the RTL IPA-GNN returns to the start of each always block after reaching its conclusion. Incorporating RTL semantics into our modeling decisions improves coverage prediction performance in some settings as compared with domain independent GNNs.

## 2.3 Gradient-based search for test generation

We now present an algorithm to generate tests to cover different cover points in an RTL design using a trained Design2Vec model. Figure 3 shows the overall flow. The key idea of Algorithm 1 is to perform a gradient based search to maximize the Design2Vec predicted probability of covering desired cover points. The algorithm takes as input a set $\mathcal{C}$ of uncovered points for which we would like

**Algorithm 1** Test Input Generation

1: Input: A set of uncovered points $\mathcal{C} = \{C_1, \cdots, C_m\}$
2: $\mathcal{I} = \{\}$
3: **while** $\mathcal{C} \neq \emptyset$ **do**
4:     $C = \texttt{PickRandomCoverPoint}(\mathcal{C})$
5:     $\mathcal{C} = \mathcal{C} \setminus C$
6:     **for** $j = 1 \dots K$ **do**
7:         {Optimize cover prob. wrt test parameters $I$}
8:         $I \leftarrow \texttt{random}()$
9:         **while** $I$ has not converged **do**
10:             $I \leftarrow I + \nabla_I \texttt{is\_hit}(C, I)$
11:         **end while**
12:         $\mathcal{I} \leftarrow \mathcal{I}.\texttt{append}(I)$
13:     **end for**
14: **end while**
15: **return** $\mathcal{I}$

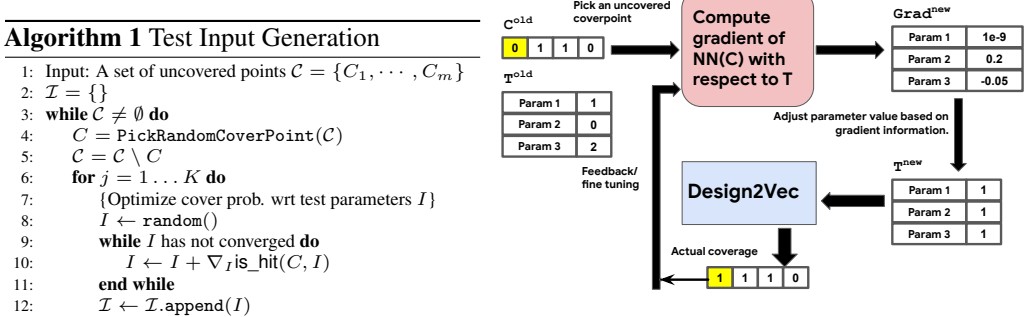

Figure 3: The overall workflow to use Design2Vec in generative loop to generate input test parameters.

to generate tests. For each uncovered point $C \in \mathcal{C}$, the algorithm first selects a random test $I$, a vector of test parameters. The algorithm then computes the objective function as the predicted probability of covering the input cover point, and computes gradients of the objective function with respect to each parameter of $I$. The test parameters in $I$ are updated using a gradient ascent method until $I$ converges (Line 10 in Algorithm 1). This process is repeated from different random initializations of $I$ to get a list of $K$ test parameters. Finally, the tests with highest predicted coverage are run through an RTL simulator to get the actual coverage.

## 3 Impact of Design2Vec solutions in practical verification

In this section, we detail the verification flow, and highlight the value proposition brought by the Design2Vec solutions to this problem. Figure 6 in the Appendix A.2 shows the industrial verification flow. The current approach for verifying RTL designs is constrained random verification [9]. This involves a complex program called the testbench. The input to the testbench is a set of test parameters written by verification engineers that define the distribution over inputs that will be applied to the RTL design. Some inputs need to be constrained, while others will be randomized by the testbench to widely sample the input space. The testbench output is a Boolean level stimulus that is applied to the primary inputs of the RTL design. Each such high level "test" corresponds to millions of cycles of Boolean input stimuli that run through an overnight regression. Bug reports and coverage are assessed at the end of a nightly regression. This process is iterated upon until there are 0 bugs and 100% of the design cover points are covered. Simulator calls are costly and need to be minimized. Effectively, this entire process costs multiple engineer years of productivity and resources.

Figure 7 illustrates how Design2Vec would integrate in the loop. We select two tasks for Design2Vec to provide practical value in constrained random verification: (i) **Coverage prediction**, or predicting which cover points in the design would be covered by a given test. Such a predictive model can serve as a **proxy simulator**, and an engineer can query it to estimate coverage instantaneously, instead of waiting for a night of simulation. (ii) **Test parameter generation**, or automating tests towards faster coverage closure, especially for covering **hard-to-cover points**. [3] in the verification cycle.

## 4 Coverage prediction experiments

We first evaluate the Design2Vec architecture on the task of coverage prediction. We evaluate two key research questions: (i) whether Design2Vec is able to exploit information from the RTL hardware design to improve predictions and (ii) whether coverage prediction is good enough for Design2Vec to serve as a proxy simulator.

We evaluate our methods on the three real-world designs from Table 6. For each design, we obtain training data by generating random tests and sampling each test parameter uniformly. For each test, we use the testbench to randomly sample input test stimulus, and a Verilog RTL simulator to obtain ground-truth labels of whether a cover point is covered by that test or not, in the form of a coverage

---

[3]In this paper we use a tool that is better than human guided random testing as a baseline for hard to cover points.

vector. We sample 1696 tests for IBEX v1, 1938 for IBEX v2, and 1995 for TPU. Some cover points are very easy to predict, e.g., they are covered by almost 0% or nearly 100% of the random tests. To avoid evaluating our models on these trivial cases, we only include cover points in the data if they are covered by between 10% and 90% of the random tests. After this filtering, there are 160 cover points in the data set for the IBEX v1 design, 177 for IBEX v2, and 3781 for TPU.

Furthermore, there is a particular type of generalization that is vital to applying coverage prediction in practice. Recall that when a design engineer uses a coverage prediction model, it is because they are proposing a new test that should exercise a cover point that is not exercised by the existing test. Therefore, in practice the primary concern is how well the model predicts whether a test that it has never seen during training will exercise a cover point *that the model has never seen how to cover during training*. For this reason, we divide the data into training and test point by cover point, such that no cover point and no test occurs in both the training and validation set for the learned models.[4] All results are the median over three random train-validation splits.

We ask two key questions in our experiments: 1) does representation learning allow Design2Vec to generalize from cover points in the training set to related cover points (e.g. neighbors in CDFG) and 2) do the graph neural networks provide deeper learning of CDFG graph structure and cover point relationships than more shallow representations?

Table 1 compares Design2Vec to three baselines across all three designs, varying the size of the training set. One is the naive statistical frequency baseline of guessing the most common value; it is the average positive rate over all cover points in the validation data (or (1-average positive rate), whichever is larger). This baseline does not take into account any correlations between cover points or test inputs. The second baseline is a multilayer perceptron (MLP) that treats the design as a black box, and does not take into account any information from the RTL. The MLP takes as input the test parameters, represented same as in Design2Vec, and the numeric index of the cover point, represented as a one-hot vector. While it cannot generalize across cover points, the MLP can still learn to generalize across test parameters (e.g., some test parameters activate many cover points, some few). In that sense it is stronger than statistical frequency.

The third baseline is node sequence embedding, a stronger baseline that enhances MLP by allowing it to generalize across cover points. Recall that every cover point is defined as a sequence of nodes down a control-flow path from the root of an always block to a particular node, e.g., `n1-n2-n4` in Figure 1. We use node sequences over all the training cover points in a Word2Vec model to learn embeddings for each control flow node, which is then concatenated and padded into a cover point embedding. This representation of the cover point is used instead of the one-hot representation in the MLP. Cover points that have structural proximity in the CDFG graph, e.g., `n1-n2-n4` and `n1-n3-n4` would have similar embeddings. This baseline can generalize to new cover points if there is a nearby coverpoint in training. While it takes graph structure into account, it does not have the full flexibility of GNN-style message propagation. First two baselines answer question 1 while node sequence embedding helps answer 3.

In these results, Design2Vec uses the RTL IPA-GNN (Section 2.2) as the GNN layer. Hyperparameters including number of layers, learning rates, and embedding dimensions are reported in Appendix B.

Notably, the MLP performs catastrophically poorly on the test data. It is slightly better than statistical frequency. Indeed, on the largest design, the industrial TPU design, Design2Vec has an accuracy of 47% higher *in absolute terms* than the MLP. We observe that the training accuracy of MLP is high (above 95%), so the MLP overfits and fails to generalize to cover points outside the training set. This is expected, since it has no information about which of the training cover points are most close to the cover points in validation set. The MLP prediction is based only on test parameter features. Node sequence embedding performs much better than the two blackbox models in every case, indicating that even shallow representation learning using RTL CDFG structure helps generalization.

The Design2Vec model wins by a substantial margin in every case, indicating that the GNN based architecture is able to learn deep relationships in the design and generalize effectively to majority of the cover points unseen during training.

---

[4] We avoid using the term "test set" to refer to the data on which we evaluate our machine learning methods, to avoid confusion with tests of the RTL design.

Table 1: Accuracy at coverage prediction for Design2Vec (with the RTL IPA-GNN layer) compared to a black-box multi-layer perceptron (MLP), which does not have information about the design.

| Train cover points | IBEX v1 | | | IBEX v2 | | | TPU | | |
| --- | --- | --- | --- | --- | --- | --- | --- | --- | --- |
| | 50% | 80% | 90% | 50% | 80% | 90% | 50% | 80% | 90% |
| Design2Vec | **74.2** | **77.3** | **77.8** | **73.4** | **78.0** | **80.3** | **90.5** | **90.6** | **91.1** |
| Node seq embedding | 59.7 | 59.1 | 63.2 | 58.5 | 57.3 | 59.0 | 87.9 | 88.4 | 88.6 |
| MLP | 57.5 | 56.8 | 56.8 | 58.7 | 58.0 | 58.2 | 42.8 | 42.5 | 34.7 |
| Statistical frequency | 50.5 | 51.6 | 50.8 | 54.1 | 54.5 | 54.7 | 68.5 | 68.6 | 68.6 |

Table 2: Coverage prediction accuracies of different GNN architectures within Design2Vec.

| | Node fusion | IBEX v1 | | | IBEX v2 | | | TPU | | |
| --- | --- | --- | --- | --- | --- | --- | --- | --- | --- |
| | | 50% | 80% | 90% | 50% | 80% | 90% | 50% | 80% | 90% |
| GCN | lstm | 74.0 | 73.0 | 73.2 | 70.4 | 74.5 | 73.9 | 90.9 | 90.5 | 91.1 |
| | mean | 74.1 | 75.8 | 74.0 | 69.0 | 73.9 | 72.5 | — | — | — |
| GGNN | lstm | **75.0** | 75.9 | 74.5 | 71.4 | 77.0 | 76.0 | **91.1** | 90.4 | **91.2** |
| | mean | 73.8 | 75.6 | 76.4 | 70.0 | 75.1 | 72.3 | — | — | — |
| GNN-MLP | lstm | 73.8 | 76.8 | **78.5** | 70.7 | 74.5 | 71.2 | 91.0 | 90.1 | 91.1 |
| | mean | 73.0 | 76.4 | 74.4 | 70.1 | 75.1 | 72.0 | — | — | — |
| RTL IPA-GNN | lstm | 74.2 | **77.3** | 77.8 | **73.4** | **78.0** | **80.3** | 90.5 | **90.6** | 91.1 |
| | mean | 73.6 | 77.2 | 76.3 | 72.2 | 75.8 | 78.9 | — | — | — |

This pattern holds across all three designs. On IBEX v1 and v2, the difference between the two models is smaller (although still around 15% absolute). This is due to the irregular structure of the Ibex designs, with fewer repeated hardware modules, causing the control flow path to be harder to predict. On the TPU industrial design is the largest and most complex of the three, Design2Vec has over 90% accuracy. The node sequence embedding also has high accuracy on this design. This can be attributed to the highly regular structure of TPU with a relatively easier to predict control flow.

For Design2Vec, we also compare several variants of graph neural networks: graph convolution networks [25], gated graph neural networks [28], GNN-MLP [7], and the RTL IPA-GNN (Section 2.2). These results are shown in Table 2. Additionally, we vary the method by which representations of CDFG nodes are aggregated to represent cover points. Recall from Section 2.1 that to represent a cover point, we take the final node embeddings from the GNN, apply a mask to obtain only a few relevant CDFG, and then aggregate them. We compare using an LSTM, as in (4), to simply taking the mean. We compare this for different GNN architectures, and note that aggregation methods performed similarly on IBEX, so we evaluate only LSTM aggregation on TPU. Overall, all GNN variants performed similarly. RTL IPA-GNN performs similarly to the other GNN architectures on IBEX v1 and TPU, but is significantly better on IBEX v2. Since the RTL IPA-GNN performs similarly or better than other GNNs across the three designs, we use this as the main Design2Vec model in Table 1. We also tuned several other features of the architecture, including presence of the residual connections, embedding size, and label smoothing (details in Appendix B and C).

**Given that TPU is a typical ML accelerator chip whose architecture will have similar properties across generations, the high coverage prediction accuracy (>90%) shows the potential of Design2Vec to serve as a proxy simulator taking seconds instead of a night of simulations.**

## 5   Test generation using Design2Vec

In this section, we evaluate Design2Vec on the test generation task, especially for hard to cover points during coverage closure. In general, there are two RTL testing approaches, namely directed testing and random testing. Directed testing refers to targeted testing of specific functionality in RTL, whereas random testing refers to undirected, random perturbation of inputs with the goal of covering the design space maximally. Design2Vec based test generation is directed and should ideally be compared with a directed testing baseline. However, we did not find a comparable directed testing tool in the open source (or commercially).[5]

---

[5]At this time, there is no practical tool for automated directed testing in industry. Some directed testing tools have been proposed in literature but are not available in the public domain for comparison. Random testing tools in open source use different tool flows and RTL language, making a comparison infeasible.

We present two comparative studies between Design2Vec and Vizier SRP, a random testing tool [18] that uses black-box optimization to maximize total coverage all cover points with every test. This comparison is inherently unequal due to the difference in their objectives of the two tools. We compare both tools with respect to coverage achieved and number of simulator calls made by each tool to achieve that coverage. Given that simulator calls are expensive, this is a relevant metric to compare.

Vizier SRP functions as follows. In each trial, it generates a test for the total set of cover points in the design, and calls the simulator. It uses actual coverage feedback in an active learning loop along with Bayesian optimization to generate tests with progressively higher coverage.

In the first study, we find points that are *to hard to cover* for Vizier SRP and challenge Design2Vec to generate tests for those points. To find hard to cover points, we run Vizier SRP until around 80% cumulative coverage is reached. From Table 3, it is seen that the remaining uncovered points are indeed rare from cover probabilities on a randomly sampled dataset (not used for training).

We intercept Vizier SRP after running for a number of trials (200 for IBEX v1, 243 for TPU). We provide the same sample tests collected by Vizier SRP as training data for Design2Vec. For each point uncovered until that point by Vizier SRP, we provide them as (unseen) target points for the Design2Vec test generation algorithm described in Table 1.

Design2Vec generates multiple test recommendations, of which we select the top ranking recommendation and call the simulator for evaluating actual coverage. If the target cover point is uncovered, we generate a different test using Design2Vec and repeat the same procedure until it is covered. For this study, we run a small number of Design2Vec tests (25). Table 7 in Appendix A demonstrates some example parameterized tests generated by Design2Vec.

We continue to run Vizier SRP further (upto 400 trials) and record the coverage at every trial (between 201-400). If an uncovered point gets covered, we measure the number of simulator calls (trials) taken by Vizier SRP to cover it for the first time and compare with the number of simulator calls (tests) used by Design2Vec to cover it. Note that Vizier SRP uses active learning for the remaining trials until 400, while Design2Vec uses zero shot learning in this configuration.

After 200 trials of Vizier, there were 22 uncovered points in IBEX and 23 uncovered points in the TPU. Table 3 shows the results of this comparison. Both Vizier SRP and Design2Vec cover 3 cover points. Vizier SRP covers 2 cover points that Design2Vec does not. Design2Vec covers 1 cover point within 12 tests that Vizier SRP does not upto 400. Similarly, for 882, due to its low probability, Vizier SRP needs 24 tests to cover it, while Design2Vec uses only 3 tests to cover. Neither Design2Vec nor Vizier SRP cover 16 cover points. We expect Design2Vec to cover more if it is run beyond 25 tests. Table 4 shows similar results on the hard to cover points in the TPU design.

In the second study, we compare Design2Vec with Vizier as a test generation tool for overall coverage. We train the two tools independently with different datasets, closer to the practical use case. We evaluate the total coverage of Design2Vec as compared to Vizier for a sample of randomly selected points with varying cover probabilities (in the spectrum of always covered to rarely/never covered). We randomly select 10 cover points from each of the three buckets of cover probabilities and hid these 30 cover points from Design2Vec. We generate tests for each of the 30 cover points using individual cover points as the search objective and report the number of tests required for Design2Vec and Vizier to hit the cover point for the first time. Table 12 in Appendix D shows the summarized observations over the 30 randomly selected points. Design2Vec is clearly valuable when generating tests hard to cover points.

In practice, the intended use case of Design2Vec is to complement Vizier SRP. While Vizier SRP can maximize for cumulative coverage, Design2Vec can generate tests for hard to cover points. Notably, hard to cover points take a lot of time and resources to cover in practice. **These results show the power of representation learning in Design2Vec. Despite the lack of examples covering hard to cover points, Design2Vec is able to generate a test to cover these cover points with orders of magnitude fewer simulator calls than a black-box optimizer and random testing.**

# 6   Related work

**Automated test generation for RTL:** Most previous techniques for RTL test generation generate Boolean level stimuli at the inputs of the RTL design [36, 31, 32, 27, 8]. By virtue of operating at

Table 3: Comparison of Design2Vec and black-box optimizer tests for covering hard to cover points: IBEX v1. Number of tests are RTL simulations.

| Cover point | | Number of used tests | |
|---|---|---|---|
| ID | Prob. | Vizier | Design2Vec |
| 401 | 0.0012 | Not covered | **29** |
| 526 | 0.0059 | Not covered | **12** |
| 528 | 0.0035 | Not covered | **10** |
| 879 | 0.0071 | 42 | **14** |
| 882 | 0.0059 | 24 | **3** |
| 886 | 0.0018 | 93 | **25** |
| 664 | 0.01 | 55 | Not covered |
| 881 | 0.0053 | 108 | Not covered |
| 14 cover points | | Not covered | Not covered |

Table 4: Comparison of Design2Vec and black-box optimization to hit target cover points: TPU. Number of tests are RTL simulations.

| Cover point | | Number of used tests | |
|---|---|---|---|
| ID | Prob. | Vizier | Design2Vec |
| 35793 | 0.0 | 22 | **17** |
| 36996 | 0.0 | 90 | **2** |
| 36372 | 0.0 | 78 | **17** |

Table 5: Comparing test generation of easy, medium and hard to cover points. Design2Vec is very efficient at generating tests for hard to cover points. Summarized result of table in Appendix Table 12.

| Cover probabilities | Difficulty | Summary of comparison |
|---|---|---|
| [0.5, 1.0) | Easy | Both Vizier and Design2Vec cover all points with a single test. |
| [0.2, 0.5) | Medium | Vizier and Design2Vec cover 9 out of 10. Design2Vec takes 3 fewer tests on average. |
| [0.05, 0.2) | **Hard** | **Design2Vec takes 20 fewer tests than Vizier on average.** Upto 40 fewer in cases. |

a higher abstraction level of parameterized tests, it is orders of magnitude more scalable than the Boolean input stimulus generation techniques. Static analysis based approaches for test generation [34, 19, 5] are based on traversing the state transition graph at a logic gate level, or the RTL design have inherent scalability limitations. Approaches that combine static and dynamic analysis like HYBRO [29] and concolic testing [36, 31, 32] in RTL rely on SMT/SAT solvers, which are also limited by scale. Other approaches random forests and decision trees with static analysis and formal verification [30, 14] require manual feature engineering and handcrafted algorithms.

**Neural program testing:** While our approach learns the semantics of a design, learning based fuzzing [17] does not take program semantics into account. In contrast to our approach that models semantics of an RTL design, Neuzz [35] approximates a program using only the input-output behavior of the program, which is similar to our black-box MLP baseline. GMETAEXP [11] models test generation as a reinforcement learning problem with the objective of maximizing total program coverage, where programs are represented using a GGNN.

**Learning to execute:** Among methods that model program semantics, [38] present an approach to use LSTM to embed a program as input and generate the output as the output sequence. IPA-GNN [6] represents the program execution using an RNN augmented with a differentiable mechanism to represent next instruction after a statement execution. Unlike these works that learn to generate program output, we tackle the problem of reachability, or learning to reach a state/node in a graph.

**Abstractions for verification:** Abstraction techniques have been used to scale design verification for many decades. Some techniques are property specific [10, 26], design specific [20], data abstractions (word level abstractions) [24, 37], language based (abstract interpretation) [12, 22], execution or structure based [13]. These abstractions are typically defined manually and are challenging to create in a precise way to both scale the verification as well as maintain desired precision. In contrast, GNN based abstractions are task-specific, in the form of continuous representations of the CDFG nodes, that generalizes both across tasks, as well as designs.

## 7 Conclusion

We present an approach that is able to learn representations of hardware designs that can predict their functionality. With Design2Vec, we demonstrate, for the first time, the ability of deep learning in creating abstractions of the hardware design state space. These abstractions outperform black-box baselines and human baselines (state of practice) in verification by orders of magnitude in performance as well as scale. They can also generalize across different designs. Since Design2Vec learns reachability over the RTL design graph, it can also potentially generalize to other verification tasks like debugging, property generation and root causing. More broadly, this work shows the power and potential of deep learning to make a quantum leap in progress in the area of verification, which is considered a longstanding practical and scientific challenge in computing.

## Acknowledgements

We thank Milad Hashemi and Sat Chatterjee for insightful discussions. We thank Dexter Aronstam and Gil Tabak for developing the key components of Design2Vec infrastructure that enabled us to perform experiments.

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
