# Supplementary Material for Learning Semantic Representations to Verify Hardware Designs

Vasudevan, Jiang, Bieber, Singh, Shajaei, Ho, Sutton, NeurIPS 2021

## Appendix A  Additional figures

### A.1  RTL CDFGs

We show an example of RTL CDFG execution (simulation) over multiple cycles in Figure 4. In the example, at a given clock cycle `t`, the values of `a` and `b` from the previous clock cycle `t-1` will be used for evaluating the condition `a > b` in the green always block and the corresponding branch will be executed in that cycle. In the other two always blocks, in cycle `t`, `b` and `a` will be assigned values based on values of `c` and `d` from previous cycle `t-1`. In the next cycle `t+1`, `a` and `b` will get values of `a` and `b` from cycle `t`.

The input stimulus and the branches covered by the simulation are shown in Figure 5.

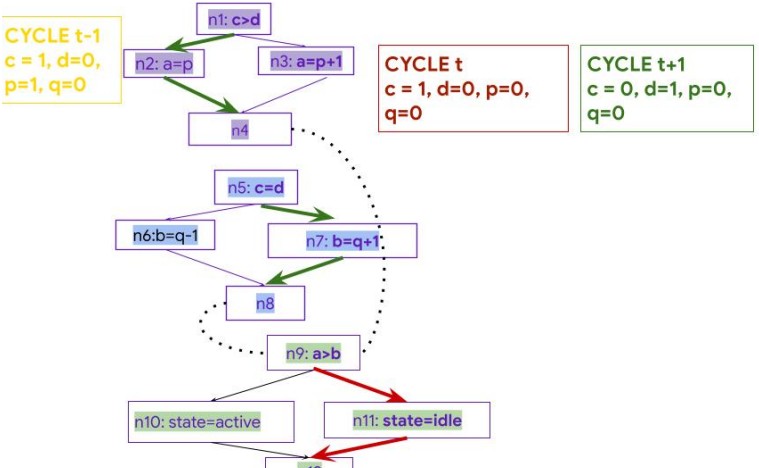

Figure 4: RTL and CDFG execution (simulation) over three cycles t-1, t, t+1

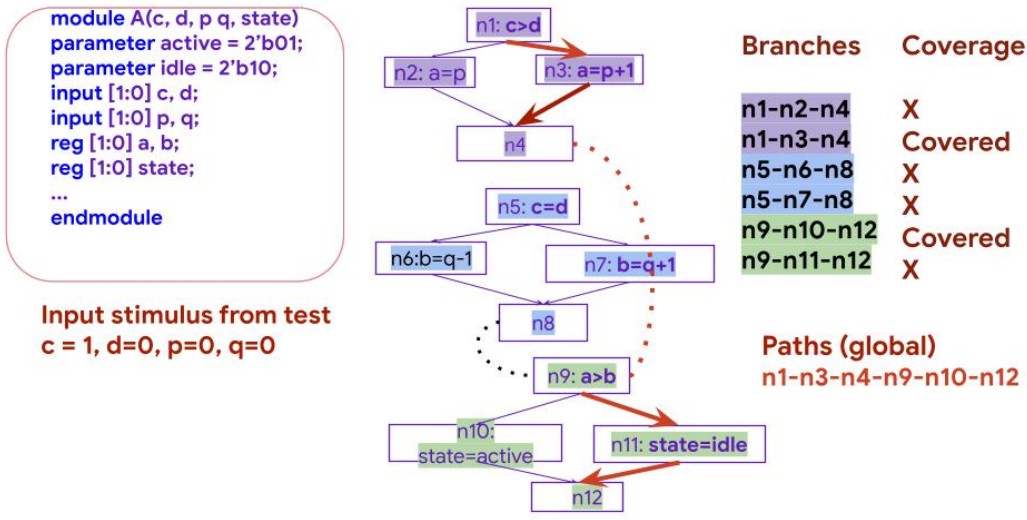

Figure 5: Input stimulus and corresponding branches that are covered. Coverage is a path tracing through the CDFG.

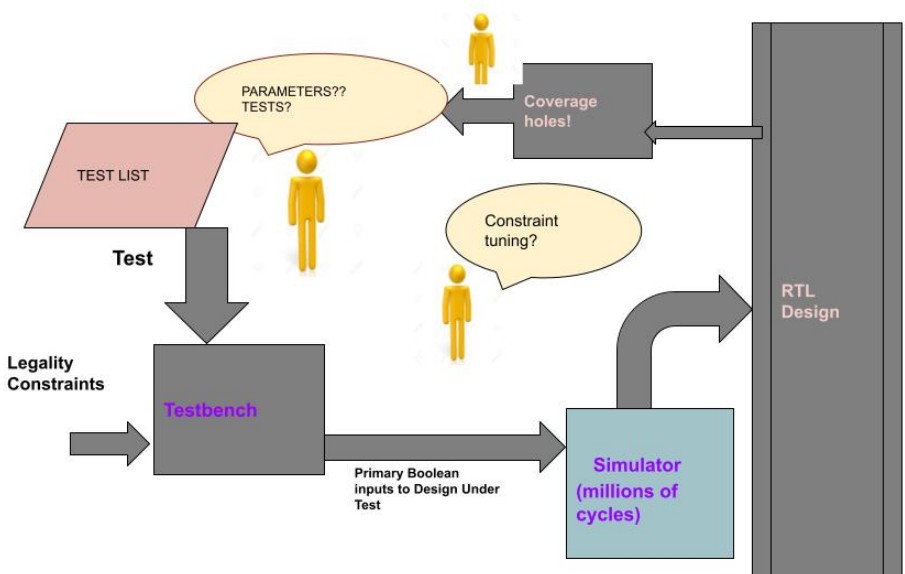

Figure 6: Industrial verification flow with manually generated testbench, tests, constraints and coverage feedback. This flow takes multiple person-years of engineer productivity to converge

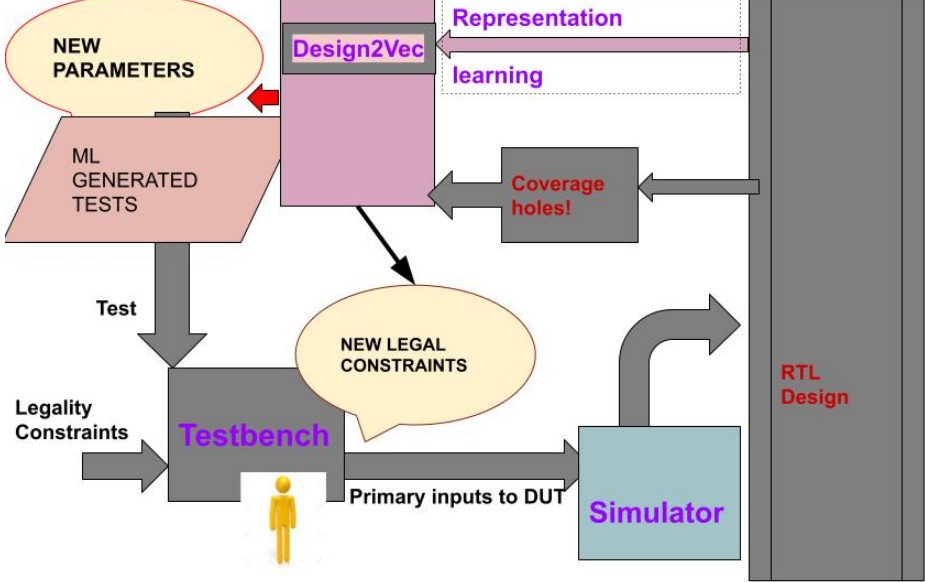

Figure 7: Value proposition of Design2Vec when integrated into the loop of an industrial verification flow. It learns about the design state space and generates tests to cover different uncovered cover points (holes). It can potentially be used to generate constraints.

## A.2  Industrial verification flow

Figure 6 shows the context of our solution within the industrial verification flow. Figure 7 shows the Design2Vec solution inbuilt into the constrained random verification environment.

## A.3  Sizes of designs

We show a comparison of the relative sizes of RTL CDFGs between IBEX and the TPU design in Table 6.

Table 6: Comparison of relative sizes of RTL CDFGs between IBEX and TPU design.

|  | IBEX v1 | TPU block |
|---|---|---|
| # CDFG Nodes | 5500 | 40000 |
| # CDFG Edges | 9300 | 58000 |
| # Branch cover points | 900 | 45000 |
| # Number of gates | 10028 | 1088343 |

Table 7: Example tests generated by Design2Vec for hard to cover points. Multiple cover points that are local neighbors of the target point are provided as input to Design2Vec, which helps the GNN-based architecture.

| Cover points | Top test recommendations |
|---|---|
| $\{521, \ldots \mathbf{526}, \ldots, 531\}$ | `+instr_cnt=17600 +illegal_instr_ratio=35 +hint_instr_ratio=45 ...`
`+enable_unaligned_load_store=0 +disable_compressed_instr=1 +randomize_csr=0 +no_wfi=1` |
| $\{881, \ldots \mathbf{882}, \ldots, 891\}$ | `+instr_cnt=17400 +illegal_instr_ratio=25 +hint_instr_ratio=35 ...`
`+enable_unaligned_load_store=0 +disable_compressed_instr=1 +randomize_csr=0 +no_wfi=1` |

### A.4 Examples of generated tests

## Appendix B  Experimental hyperparameters

The methods in Section 4 use the following hyperparameters: number of layers, learning rate, GNN embedding dimension, residual connection frequency, dropout rate, MLP embedding dimensions. We vary these parameters: number of layers $\in \{4, 8, 16, 32\}$, learning rate $\in \{1e-2, 1e-3, 3e-4, 1e-4\}$. We hole these parameters fixed: GNN embedding dimension $= 16$, residual connection frequency $= 4$, dropout rate $= 0.1$, MLP embedding dimensions $= [256, 128, 64]$. The IPA-GNN model only has one additional hyperparameter: normalization term $\in \{1, p_{t,n}\}$.

Number of layers describes the number of GNN layers in the Design2Vec model. GNN embedding dimension describes the embedding dimension of the intermediate and final node embeddings produced by the GNN. The residual connection frequency indicates where skip connections are added between layers of the GNN. The normalization term is used to normalize $h_{t,n}$ after each RTL IPA-GNN layer.

Our experiments were run on commodity GPUs in a commercial data center. In total, the experiments reported here required approximately five GPU-weeks of computation.

## Appendix C  Architectural ablation studies

In this section, we present more detailed comparison of different hyperparameter settings and ablations of Design2Vec. Since we hypothesize that propagating information across longer distances in the graph is important, we were especially interested in the effect of residual connections [21], but we also measure the effect of label smoothing [33]. See results in Table 8. Our default Design2Vec architecture (top row in table) uses residual connections that skip back 4 layers, and does not use label smoothing. In practice, we see that none of these variants have a large effect on performance.

Table 8: Comparison of training and validation accuracy across variants of the Design2Vec architecture on coverage prediction on the TPU design.

|  | Training | Validation |
|---|---|---|
| Design2Vec (GCN) | 92.1 | 90.3 |
| Design2Vec + no residual connections | 92.1 | 90.7 |
| Design2Vec + label smoothing | 92.1 | 90.3 |
| Design2Vec + residual (every 2 layers) | 92.1 | 90.1 |

We further consider architectural ablations that vary the K-hop edges added to the input CDFG in Table 11, as well as the depth of the network in Table 10. We further test each setup in settings that vary the selection of training and validation splits in Tables 9.

Table 9: Comparing the train and validation accuracy across different split selection methods: whether to hide test parameters, and whether to sample the training set via every-k sampling or uniformly random cover points.

| Hide test params | Cover point hiding method | Seed | Training | Validation |
|---|---|---|---|---|
| FALSE | Deterministic | — | 86.97 | 84.54 |
| FALSE | Random cover point | 123 | 87.41 | 76.86 |
| FALSE | Deterministic | — | 82.47 | 84.45 |
| FALSE | Random cover point | 123 | 80.09 | 75.93 |
| FALSE | Deterministic | — | 87.80 | 80.76 |
| FALSE | Random cover point | 123 | 87.73 | 80.51 |
| FALSE | Deterministic | — | 83.09 | 79.62 |
| FALSE | Random cover point | 123 | 83.18 | 78.80 |
| FALSE | Deterministic | — | 89.91 | 80.58 |
| FALSE | Random cover point | 123 | 89.53 | 78.06 |
| FALSE | Deterministic | — | 78.62 | 76.34 |
| FALSE | Random cover point | 123 | 83.39 | 75.08 |
| TRUE | Deterministic | — | 87.17 | 82.67 |
| TRUE | Random cover point | 123 | 87.51 | 75.92 |
| TRUE | Deterministic | — | 82.50 | 83.97 |
| TRUE | Random cover point | 123 | 82.83 | 78.51 |
| TRUE | Deterministic | — | 88.07 | 78.32 |
| TRUE | Random cover point | 123 | 87.96 | 75.98 |
| TRUE | Deterministic | — | 83.28 | 78.23 |
| TRUE | Random cover point | 123 | 83.40 | 78.06 |
| TRUE | Deterministic | — | 90.16 | 77.27 |
| TRUE | Random cover point | 123 | 89.78 | 73.84 |
| TRUE | Deterministic | — | 62.89 | 63.79 |
| TRUE | Random cover point | 123 | 83.51 | 73.63 |

Table 10: Comparing the train and validation accuracy on TPU while varying the numbers of GCN layers. We report the results across three seeds for each network depth.

| Seed | GCN layers | Training | Validation |
|---|---|---|---|
| 111 | 3 | 92.13 | 90.75 |
| 123 | 3 | 92.13 | 90.13 |
| 321 | 3 | 92.11 | 90.67 |
| 111 | 12 | 92.12 | 90.53 |
| 123 | 12 | 92.14 | 90.21 |
| 321 | 12 | 92.12 | 90.61 |
| 111 | 24 | 92.11 | 90.19 |
| 123 | 24 | 92.14 | 90.82 |
| 321 | 24 | 92.09 | 90.55 |

Table 11: Comparing the training and validation accuracy of the Design2Vec model using a k-hop edge augmented graph ($k \in \{2, 4, 16\}$) across a variety of experimental setups.

| K-hop | Hide test params | Cover point hiding method | Seed | Training | Validation |
|---|---|---|---|---|---|
| 2 | FALSE | Deterministic | — | 86.88 | 83.48 |
| 2 | FALSE | Random cover point | 123 | 87.25 | 76.87 |
| 2 | FALSE | Deterministic | — | 65.07 | 69.14 |
| 2 | FALSE | Random cover point | 123 | 82.47 | 74.43 |
| 2 | FALSE | Deterministic | — | 87.62 | 79.08 |
| 2 | FALSE | Random cover point | 123 | 87.68 | 77.02 |
| 2 | FALSE | Deterministic | — | 82.66 | 78.34 |
| 2 | FALSE | Random cover point | 123 | 66.42 | 61.58 |
| 2 | FALSE | Deterministic | — | 89.81 | 78.92 |
| 2 | FALSE | Random cover point | 123 | 89.26 | 77.17 |
| 2 | FALSE | Deterministic | — | 84.25 | 79.11 |
| 2 | FALSE | Random cover point | 123 | 84.69 | 74.36 |
| 2 | TRUE | Deterministic | — | 87.00 | 82.30 |
| 2 | TRUE | Random cover point | 123 | 87.39 | 75.69 |
| 2 | TRUE | Deterministic | — | 83.15 | 83.04 |
| 2 | TRUE | Random cover point | 123 | 82.87 | 72.78 |
| 2 | TRUE | Deterministic | — | 87.74 | 76.93 |
| 2 | TRUE | Random cover point | 123 | 87.83 | 76.19 |
| 2 | TRUE | Deterministic | — | 82.83 | 75.75 |
| 2 | TRUE | Random cover point | 123 | 83.29 | 76.86 |
| 2 | TRUE | Deterministic | — | 90.04 | 76.16 |
| 2 | TRUE | Random cover point | 123 | 89.62 | 73.33 |
| 2 | TRUE | Deterministic | — | 84.86 | 76.69 |
| 2 | TRUE | Random cover point | 123 | 84.79 | 73.12 |
| 4 | FALSE | Deterministic | — | 86.86 | 85.30 |
| 4 | FALSE | Random cover point | 123 | 87.22 | 77.75 |
| 4 | FALSE | Deterministic | — | 65.03 | 69.03 |
| 4 | FALSE | Random cover point | 123 | 65.90 | 64.28 |
| 4 | FALSE | Deterministic | — | 87.63 | 80.32 |
| 4 | FALSE | Random cover point | 123 | 87.68 | 78.85 |
| 4 | FALSE | Deterministic | — | 82.83 | 79.74 |
| 4 | FALSE | Random cover point | 123 | 82.22 | 80.57 |
| 4 | FALSE | Deterministic | — | 89.75 | 79.98 |
| 4 | FALSE | Random cover point | 123 | 89.30 | 78.28 |
| 4 | FALSE | Deterministic | — | 84.65 | 80.27 |
| 4 | FALSE | Random cover point | 123 | 84.55 | 74.76 |
| 4 | TRUE | Deterministic | — | 87.03 | 80.18 |
| 4 | TRUE | Random cover point | 123 | 87.39 | 75.28 |
| 4 | TRUE | Deterministic | — | 82.75 | 83.00 |
| 4 | TRUE | Random cover point | 123 | 82.20 | 74.67 |
| 4 | TRUE | Deterministic | — | 87.94 | 76.56 |
| 4 | TRUE | Random cover point | 123 | 87.79 | 77.02 |
| 4 | TRUE | Deterministic | — | 83.05 | 76.05 |
| 4 | TRUE | Random cover point | 123 | 83.35 | 75.64 |
| 4 | TRUE | Deterministic | — | 90.01 | 77.08 |
| 4 | TRUE | Random cover point | 123 | 89.26 | 74.03 |
| 4 | TRUE | Deterministic | — | 81.17 | 77.00 |
| 4 | TRUE | Random cover point | 123 | 84.70 | 73.24 |
| 16 | FALSE | Deterministic | — | 86.97 | 84.39 |
| 16 | FALSE | Random cover point | 123 | 87.30 | 73.10 |
| 16 | FALSE | Deterministic | — | 64.92 | 68.95 |
| 16 | FALSE | Random cover point | 123 | 82.62 | 76.84 |
| 16 | FALSE | Deterministic | — | 87.67 | 79.56 |

| 16 | FALSE | Random cover point | 123 | 87.66 | 76.26 |
| 16 | FALSE | Deterministic | — | 64.94 | 65.32 |
| 16 | FALSE | Random cover point | 123 | 83.14 | 78.83 |
| 16 | FALSE | Deterministic | — | 89.79 | 79.80 |
| 16 | FALSE | Random cover point | 123 | 89.37 | 77.98 |
| 16 | FALSE | Deterministic | — | 84.95 | 77.44 |
| 16 | FALSE | Random cover point | 123 | 84.64 | 73.66 |
| 16 | TRUE | Deterministic | — | 87.04 | 79.44 |
| 16 | TRUE | Random cover point | 123 | 87.41 | 73.71 |
| 16 | TRUE | Deterministic | — | 64.57 | 66.58 |
| 16 | TRUE | Random cover point | 123 | 82.43 | 73.24 |
| 16 | TRUE | Deterministic | — | 87.85 | 75.15 |
| 16 | TRUE | Random cover point | 123 | 87.85 | 73.63 |
| 16 | TRUE | Deterministic | — | 83.45 | 77.44 |
| 16 | TRUE | Random cover point | 123 | 66.86 | 62.25 |
| 16 | TRUE | Deterministic | — | 90.16 | 76.61 |
| 16 | TRUE | Random cover point | 123 | 89.65 | 72.47 |
| 16 | TRUE | Deterministic | — | 84.25 | 77.18 |
| 16 | TRUE | Random cover point | 123 | 84.57 | 70.69 |

## Appendix D    Comparison of Design2Vec and black-box optimizer tests for covering overall cover points

Table 12: Comparison of Design2Vec and black-box optimizer tests for covering overall cover points in different cover probability buckets.

| Cover Prob. Bucket | Cover Point ID | Cover Prob. | Covered by Design2Vec? | # Design2Vec tests | Covered by Vizier? | # Vizier tests | # D2V - # Vizier |
|---|---|---|---|---|---|---|---|
| $[0.5, 1.0)$ | 97 | 99.94% | Yes | 1 | Yes | 1 | 0 |
| $[0.5, 1.0)$ | 113 | 99.94% | Yes | 2 | Yes | 1 | 1 |
| $[0.5, 1.0)$ | 158 | 87.85% | Yes | 1 | Yes | 1 | 0 |
| $[0.5, 1.0)$ | 164 | 53.36% | Yes | 3 | Yes | 1 | 2 |
| $[0.5, 1.0)$ | 266 | 98.70% | Yes | 1 | Yes | 1 | 0 |
| $[0.5, 1.0)$ | 394 | 98.00% | Yes | 1 | Yes | 1 | 0 |
| $[0.5, 1.0)$ | 810 | 84.79% | Yes | 1 | Yes | 1 | 0 |
| $[0.5, 1.0)$ | 841 | 97.29% | Yes | 1 | Yes | 1 | 0 |
| $[0.5, 1.0)$ | 850 | 96.93% | Yes | 1 | Yes | 1 | 0 |
| $[0.5, 1.0)$ | 858 | 85.14% | Yes | 1 | Yes | 1 | 0 |
| $[0.2, 0.5)$ | 16 | 35.26% | Yes | 1 | Yes | 2 | -1 |
| $[0.2, 0.5)$ | 47 | 24.12% | Yes | 1 | Yes | 4 | -3 |
| $[0.2, 0.5)$ | 50 | 24.00% | Yes | 1 | Yes | 4 | -3 |
| $[0.2, 0.5)$ | 185 | 34.43% | Yes | 2 | Yes | 5 | -3 |
| $[0.2, 0.5)$ | 356 | 49.17% | Yes | 2 | Yes | 3 | -1 |
| $[0.2, 0.5)$ | 422 | 49.17% | Yes | 4 | Yes | 3 | 1 |
| $[0.2, 0.5)$ | 813 | 48.58% | Yes | 1 | Yes | 5 | -4 |
| $[0.2, 0.5)$ | 816 | 48.53% | Yes | 1 | Yes | 5 | -4 |
| $[0.2, 0.5)$ | 817 | 28.83% | Yes | 1 | Yes | 9 | -8 |
| $[0.2, 0.5)$ | 818 | 38.38% | Yes | 1 | Yes | 5 | -4 |
| $[0.05, 0.2)$ | 400 | 9.38% | Yes | 5 | Yes | 2 | 3 |
| $[0.05, 0.2)$ | 506 | 7.72% | Yes | 5 | Yes | 4 | 1 |
| $[0.05, 0.2)$ | 624 | 10.02% | Yes | 1 | Yes | 13 | -12 |
| $[0.05, 0.2)$ | 646 | 6.90% | Yes | 1 | Yes | 45 | -44 |
| $[0.05, 0.2)$ | 649 | 6.72% | Yes | 5 | Yes | 45 | -40 |
| $[0.05, 0.2)$ | 656 | 5.37% | Yes | 1 | No | — | NA |
| $[0.05, 0.2)$ | 667 | 19.16% | No | — | Yes | 20 | NA |

| [0.05, 0.2) | 677 | 6.43% | Yes | 6 | Yes | 36 | -30 |
| [0.05, 0.2) | 700 | 6.43% | Yes | 5 | Yes | 36 | -31 |
| [0.05, 0.2) | 708 | 8.37% | Yes | 2 | Yes | 4 | -2 |