# OpenReview forum: "Learning Semantic Representations to Verify Hardware Designs"
_NeurIPS.cc/2021/Conference — NeurIPS 2021 Poster_

### Official Review · Reviewer_z6UE · 2021-07-16

**Rating:** 7
**Confidence:** 5

**Summary:**

This paper introduces Design2Vec, which models the semantics of RTL programs using a graph neural network. The representation is used for 2 tasks: coverage prediction and test generation. The experiments on different designs show that Design2Vec can achieve high accuracy on the coverage prediction task and can also generate tests that cover hard-to-cover points.

**Limitations And Societal Impact:**

While the authors discuss the general challenges in hardware verification, they apply mostly to formal verification. It would be beneficial if the authors could discuss challenges specific to test-based methods.

**Main Review:**

The paper is well-written. The use of learning for the particular task is well-motivated. The architecture of the network appears reasonable and the performance of the model on the two tasks shows a clear benefit of the system. Using a gradient-based method on the learned model to generate tests for some targeted cover points is a clever idea. And it is encouraging to see that this approach can actually generate meaningful tests. There seems to be real potential to deploy Design2Vec in real-world hardware testing scenarios.

The experimental evaluation is extensive and shows the benefit of the proposed architectures. The only reason that I reserve my score to 6, for now, is that I have two concerns for the second experiment (section 5). First, why do the authors choose Vizier as the baseline instead of existing RTL test generation techniques as mentioned in the first paragraph of the related work? The paper mentions that they are less scalable than the presented technique. Concrete evaluation results of some of those tools on the same designs would strengthen the claim and also provide more validation to the presented learning-based technique. Second, it is not explicit from the paper whether Table 3 and Table 4 contain all the cover points in C_uncov. If this is not the case, are there cover points whether Vizier can cover faster than Design2Vec? If so how many? Are there points that Design2Vec cannot generate tests to cover?

Minor comments:
line 99: extra 0 at the end of the line.
line 335: blacklbox -> blackbox


**Time Spent Reviewing:**

8

---

> ### Author Response · Authors · 2021-08-12
> **Response to Reviewer z6UE**
>
> Thanks for your review. We will try to address your two main concerns. Please let us know if we have not addressed them fully, and we will try to provide more detail.
>
> ## Q1. First, why do the authors choose Vizier as the baseline instead of existing RTL test generation techniques as mentioned in the first paragraph of the related work?
>
> Thanks for asking this question. Please see the common response ("comparison with other test generation tools"). We will clarify this in the paper as well.
>
> ## Q2. Second, it is not explicit from the paper whether Table 3 and Table 4 contain all the cover points in $C_\text{uncov}$. If this is not the case, are there cover points whether Vizier can cover faster than Design2Vec? If so how many? Are there points that Design2Vec cannot generate tests to cover?
>
> After 200 trials of Vizier SRP, there are 22 cover points in $C_\text{uncov}$ for IBEX and 23 in $C_\text{uncov}$ for FastML. Table 3 shows 4 of the cover points that were covered by Design2Vec from $C_\text{uncov}$ in IBEX in 22 tests (simulator calls). It takes Vizier SRP an additional 200 trials (iterations or simulator calls) to cover 5/22 cover points in $C_\text{uncov}$ for IBEX. We did not try more than 22 tests from Design2Vec for that study.
>
> Both Vizier SRP and Design2Vec cover: 3 CPs.
>
> Vizier SRP covers, but not Design2Vec: 2 CPs.
>
> Design2Vec covers, but not Vizier SRP: 1 CP.
>
> Not covered by both Design2Vec and Vizier SRP: 22 - 3 - 2 - 1 = 16 CPs.
>
> For the remaining 16 CPs that were not covered by either in our study, we are running experiments with more tests from Design2Vec and Vizier SRP to get the analysis you have suggested.
>
> Table 4 shows 3 of the cover points that were covered by Design2Vec from $C_\text{uncov}$ in FastML. We ran only 20 tests (simulator calls) from Design2Vec, which covered 3 of the 23 CPs. It takes Vizier another 200 trials (iterations or simulator calls) to cover the rest of 23 CPs in $C_\text{uncov}$ for FastML. We are running experiments to find the number of tests it takes Design2Vec to cover the remaining 20 CPs.
>
> It should be noted, however, that Vizier SRP has the advantage of feedback driven active learning from 200 new samples. In its current form, Design2Vec test generation is effectively zero-shot and does not include active learning. As such the test generation in the current state of Design2Vec is not a strictly fair comparison to Vizier SRP's test generation.
>
> The intended use case of Design2Vec is not to replace Vizier SRP, but to be complementary to it. The SRP generated tests and Design2Vec generated tests can run in the same environment as Fig 6 in the paper, replacing the human in the loop for a majority of the cover points. While Vizier SRP can maximize for cumulative coverage and converge quickly on easy to cover points, Design2Vec can generate tests for hard to cover points. As we have shown in the paper, even with zero shot learning, there are hard to cover points in Ibex and FastML that Design2Vec can cover faster than Vizier SRP or random.
>
> Inspired by your question, we are also running another experiment for evaluating the total coverage of Design2Vec as compared to Vizier for a sample of randomly selected points with varying cover probabilities (not only the ones that are hard to cover). For this experiment, we are training a Design2Vec model offline with randomly generated samples. We will report results subsequently.

---

> > ### Author Response · Authors · 2021-08-20
> > **Additional comparison with Vizier on random cover points, and training Design2Vec on randomly sample dataset**
> >
> > Thanks again for your comments and suggestions. As we mentioned in our earlier response, we have run more experiments to address your second question (overall comparison of Design2Vec vs Vizier coverage).
> >
> > In the paper, we presented results on Ibex for points that were hard to cover by Vizier (within 200 iterations). We trained Design2Vec using the same 200 tests that Vizier had trained on. We then queried Design2Vec with the points that were as yet uncovered by Vizier. We let Vizier run for 200 more iterations. We then compared the number of tests it took Design2Vec to cover the point against the number of iterations between 201-400 that it took Vizier to cover that point. Note that Design2Vec does not have the benefit of active learning and feedback from examples, while Vizier continues to use active learning between 201-400 iterations. Also, please note that the intended use of Design2Vec is not as a substitute for Vizier, but as a complementary tool for Vizier that can be used for directed test generation that targets hard to cover points.
> >
> > While we performed the study for an academic comparison, in a practical setting, Design2Vec would not be trained on the Vizier generated (biased) samples. It would use a randomly generated set of samples from an independent dataset collected by the existing production verification process. As such, this second experiment addresses the spirit of your question better, i.e., how does Design2Vec compare overall as a test generation tool with Vizier. We evaluated the total coverage of Design2Vec as compared to Vizier for a sample of randomly selected points with varying cover probabilities (in the spectrum of always covered to rarely/never covered).
> >
> > 1) We generate tests using Design2Vec for cover points randomly selected from different cover probability buckets (indicating their ease/hardness to cover). We randomly selected 10 cover points from each bucket of cover probabilities as follows. The numbers on the right indicate cover point indices with the IBEX RTL design (1-900 for 900 cover points).
> >
> > [0.5, 1.0): [97, 113, 158, 164, 266, 394, 810, 841, 850, 858]
> >
> > [0.2, 0.5): [16, 47, 50, 185, 356, 422, 813, 816, 817, 818]
> >
> > [0.05, 0.2): [400, 506, 624, 646, 649, 656, 667, 677, 700, 708]
> >
> >
> > We trained a Design2Vec model with a randomly sampled dataset  and hid the 30 cover points above. We generated tests for each of the 30 cover points using individual cover points as the search objective. We report the number of tests required for Design2Vec and Vizier to hit the cover point for the first time.
> > Observations:
> > - Both Design2Vec and Vizier can cover all cover points in the [0.05, 0.2) bucket, except for 2 cover points. Of those, one is covered only by Design2Vec and the other is covered by Vizier.
> > - For easy [0.5, 1.0) cover points, both Design2Vec and Vizier mostly hit them with one single test.
> > - For medium difficulty [0.2, 0.5) cover points, Design2Vec uses on average 3 fewer tests than Vizier.
> > - For rare [0.05, 0.2) cover points, Design2Vec uses on average 20 fewer tests than Vizier.
> >
> >
> >
> > |Cover Probability Bucket | Cover Point Index | Cover Probability | Covered by Design2Vec? |Number of tests Design2Vec uses | Covered by Vizier? | Number of tests Vizier uses | \# Design2Vec tests - \# Vizier tests |
> > | --- | --- | --- | --- | --- | --- | --- | --- |
> > | [0.5, 1.0) |97 | 0.9994 | Yes |1|Yes|1|0|
> > |[0.5, 1.0)|113|0.9994|Yes|2|Yes|1|1|
> > |[0.5, 1.0)|158|0.8785|Yes|1|Yes|1|0|
> > |[0.5, 1.0)|164|0.5336|Yes|3|Yes|1|2|
> > |[0.5, 1.0)|266|0.987|Yes|1|Yes|1|0|
> > |[0.5, 1.0)|394|0.98|Yes|1|Yes|1|0|
> > |[0.5, 1.0)|810|0.8479|Yes|1|Yes|1|0|
> > |[0.5, 1.0)|841|0.9729|Yes|1|Yes|1|0|
> > |[0.5, 1.0)|850|0.9693|Yes|1|Yes|1|0|
> > |[0.5, 1.0)|858|0.8514|Yes|1|Yes|1|0|
> > |[0.2, 0.5)|16|0.3526|Yes|1|Yes|2|-1|
> > |[0.2, 0.5)|47|0.2412|Yes|1|Yes|4|-3|
> > |[0.2, 0.5)|50|0.24|Yes|1|Yes|4|-3|
> > |[0.2, 0.5)|185|0.3443|Yes|2|Yes|5|-3|
> > |[0.2, 0.5)|356|0.4917|Yes|2|Yes|3|-1|
> > |[0.2, 0.5)|422|0.4917|Yes|4|Yes||3|1|
> > |[0.2, 0.5)|813|0.4858|Yes|1|Yes|5|-4|
> > |[0.2, 0.5)|816|0.4853|Yes|1|Yes|5|-4|
> > |[0.2, 0.5)|817|0.2883|Yes|1|Yes|9|-8|
> > |[0.2, 0.5)|818|0.3838|Yes|1|Yes|5|-4|
> > |[0.05, 0.2)|400|0.0938|Yes|5|Yes|2|3|
> > |[0.05, 0.2)|506|0.0772|Yes|5|Yes|4|1|
> > |[0.05, 0.2)|624|0.1002|Yes|1|Yes|13|-12|
> > |[0.05, 0.2)|646|0.069|Yes|1|Yes|45|-44|
> > |[0.05, 0.2)|649|0.0672|Yes|5|Yes|45|-40|
> > |[0.05, 0.2)|656|0.0537|Yes|1|No|NA|NA|
> > |[0.05, 0.2)|667|0.1916|No|NA|Yes|20|NA|
> > |[0.05, 0.2)|677|0.0643|Yes|6|Yes|36|-30|
> > |[0.05, 0.2)|700|0.0643|Yes|5|Yes|36|-31|
> > |[0.05, 0.2)|708|0.0837|Yes|2|Yes|4|-2|
> >
> >
> >
> >
> > 2) Updated experimental results for hard to cover points
> >
> > In our first response, we presented results for hard to cover points by training Design2Vec on Vizier samples. Here, we provide the same comparative results when Design2Vec is trained on an independent dataset intended in practical settings. We expect that with the inclusion of active learning with actual coverage feedback for each test from the simulator, the number of hard to cover points hit by Design2Vec will be higher.
> >
> > Out of 22 unseen cover points
> >
> > | | \# Cover points|
> > | --- | --- |
> > |Covered by both Vizier and Design2Vec (Deisng2Vec uses much fewer tests)|3 [879, 882, 886]|
> > |Covered by Vizier only |2 [664, 881]|
> > |Covered by Design2Vec only|3 [401, 526, 528]|
> > |Neither Vizier nor Design2Vec covers|14 (remaining)|

---

> > > ### Comment · Reviewer_z6UE · 2021-08-25
> > > **Follow up**
> > >
> > > Dear authors,
> > >
> > > Thanks for the response and the additional experimental results. I believe they have addressed my concerns and suggest that Design2Vec is a viable practical alternative to Vizier. And I have updated my score to 7. For completeness of exposition, it would be good to also report cover points uniquely covered by Vizier in the revised version of the paper.

---

### Official Review · Reviewer_N6sS · 2021-07-16

**Rating:** 7
**Confidence:** 3

**Summary:**

This paper introduces a graph based deep architecture that can learn and understand hardware design. Experiment results suggest good performance in generating test for hardware verification.


**Limitations And Societal Impact:**

This paper does not discuss its limitation. Here are some of my questions and suggestions:
	1. Does the proposed method perform better in pure combinational logic (without register), it seems it may be much easier to model without state related registers, it would be interesting to see a comparison between sequential  design and combinational design.
	2. How does it scale? What would be the sweet spot in terms of  design complexity to train it on?
	3. Is possible to extend this to analog model, probably at least system-verilog.
	4. For related work, it would be great to separate them into non-ML based and ML-based, and have a section to state the novelty of the proposed method.

**Main Review:**

This paper presents a novel model Design2Vec to embed RTL design with a graph based architecture. This paper is well organised and have enough information for readers to follow. The authors provided illustrative figures to go through both RTL architecture examples and network architecture. Experimental results do show much better performance in Design2Vec compare to vanilla network in  verification coverage, and good performance for proposed RTL IPA_GNN. Overall, this is an interesting paper tacking an important problem in hardware verification.


**Time Spent Reviewing:**

3

---

> ### Author Response · Authors · 2021-08-12
> **Response to Reviewer N6sS**
>
> Thanks for your review and for your great suggestions. We will incorporate these points into the paper.
>
> ## Q1. Does the proposed method perform better in pure combinational logic (without register), it seems it may be much easier to model without state related registers, it would be interesting to see a comparison between sequential design and combinational design?
>
> Our method is based on a control data flow graph extracted from the behavioral level RTLsource code of the design. A combinational block is an always block without the clock triggers, similar to an assign statement. We allow for these constructs to be parsed and analyzed in our CDFG. The combinational or sequential nature of the design is not explicitly modeled at the gate level as combinational gates or registers with feedback loops.
>
> In our designs there are a few combinational parts and many sequential parts. Since typically control path (sequential parts) are harder to verify than combinational blocks, the interesting cover points tend to be around sequential parts of the design. At the block level, instruction sequences that trigger sequential behavior are more practically challenging than testing individual combinational parts.
>
> ## Q2. How does it scale? What would be the sweet spot in terms of design complexity to train it on?
>
> In our experiments, we have shown our technique can scale well from designs consisting of 5500 CDFG nodes (IBEX) to 40000 nodes (FastML). Modern ML infrastructure can scale to training very large models (for large CDFG) with distributed learning. In terms of design complexity, we have tried some complex blocks in an industrial scale ML accelerator design. For blocks that are data path centric, the structure tends to be relatively regular and compositional techniques may be used to instantiate regular structures. For control centric blocks like the one in this paper, there is much less regularity in the structure, making it harder to scale the GCN based models. We are planning to use transfer learning between different evolving versions of RTL, as well as blocks for common design structures like FIFO, buffer and other design library constructs.
>
> ## Q3. Is possible to extend this to analog model, probably at least system-verilog.
>
> In analog designs, the state of analog verification and test is still intensely manual and custom crafted. The Verilog AMS simulator itself is relatively recent. Analog test generation, although nascent, presents exciting challenges to neural network based test generation. It seems plausible for neural network based test generation to work with analog designs, since in prior art, ML based approaches like Variational Bayesian approaches work very well (Ahmadyan et al, Automated Transient Input Stimuli Genration for Analog Circuits, IEEE TCAD 2016).
>
> Our approach works on System Verilog at this time. All our designs are in System Verilog.
>
> ## Q4. For related work, it would be great to separate them into non-ML based and ML-based, and have a section to state the novelty of the proposed method.
>
> Thanks. We will restructure the related work accordingly.

---

### Official Review · Reviewer_D7qc · 2021-07-29

**Rating:** 6
**Confidence:** 3

**Summary:**

The paper proposes an approach for predicting whether a given set of inputs will induce a given branch to be taken in an RTL program, along with an algorithm for synthesizing inputs to the program that induce a given branch to be taken. The objective is to aid verification of proposed hardware designs by reducing the number of actual simulations required to find test cases that fully cover the control flow graph of the RTL program. The approach is to use a graph neural network, enhanced with some task-specific features, along with a gradient-descent algorithm for finding inputs that induce a given branch to be taken. The paper evaluates the approach, showing that it leads to relatively high branch-taken predictive accuracy on three hardware architectures, and showing that the approach requires fewer evaluations than Bayesian Optimization driven search to find inputs that induce a given branch.

**Ethical Concerns:**

There are no significant ethical concerns raised by the paper.

**Limitations And Societal Impact:**

There are no significant limitations or potential for negative societal impact that are immediately apparent that the authors don't address.

**Main Review:**

As context on my expertise, I am familiar with the semantics of RTL though not with all of the terminology in the paper nor deeply with the goal of design verification process of hardware. I am more of an expert in using neural networks to learn semantics of programs.

## Originality

To my knowledge, the paper is original in its approach and domain of application. The most similar approach in the literature is NEUZZ [34], which uses a similar approach for fuzz testing of programs; however, the approach and domain of application in this paper are substantially novel.

## Quality

- Section 2:
  - This comment does not affect my score, but I'm curious about the rationale behind the choice to include the test inputs at the final MLP rather than earlier in the GNN. Intuitively, I would expect it to be much easier to predict which branches are taken using a GNN that considers the program input along with the program structure, as opposed to a MLP that considers the program inputs and an embedding of the program structure.
  - The RTL IPA-GNN is claimed as a contribution, but a formal description is not in the main body of the paper. There is also no ablation to compare it to the original IPA-GNN.
  - There are not enough details provided on Algorithm 1 / Figure 3 to reproduce this work. Specifically, Figure 3 shows a step that says to "Adjust parameter value based on gradient information", but doesn't say how to actually do that. Figure 3 also says to "Pick an uncovered coverpoint", but this does not correspond to anything in Algorithm 1.
- Section 4:
  - Does not evaluate research question (ii), "whether the performance of Design2Vec can add practical value to industrial-scale designs", other than stating that 90% accuracy is relatively high (lines 251-254). It is not obvious that 90% accuracy is sufficient, or even that this specific task (predicting whether branches are taken) adds "practical value".
  - MLP baseline: maybe I misunderstand something about this, but if the input is a one-hot cover point that has never been observed in training, shouldn't the MLP essentially be no better than guessing based on frequency of whether arbitrary cover points are taken or not (by symmetry)? Given that this is the case, including the cover points into the MLP model is essentially guaranteed to lower the test accuracy, as it is noise that can be overfit on during the training phase. It is therefore hard to reason about Design2Vec's accuracy in comparison to this unfairly bad baseline.
- Section 5:
  - Despite the qualification on line 288 that the "comparison is inherently unequal", this section repeatedly compares Vizier and Design2Vec, stating that Design2Vec is superior (by bolding Table 3 and 4, comparing the number of tests required in line 310, etc.). Overall, this evaluation seems backwards to me, since it is not possible to reason about Design2Vec's performance over a flawed baseline. Instead, it might make sense to try to use Design2Vec to perform the same task as Vizier, to evaluate the hypothesis that the GNN-based surrogate model which is then optimized against using the gradients outperforms the Gaussian Process underlying Vizier which is optimized with Bayesian Optimziation; the fact that Design2Vec is able to target specific cover points seems like an additional benefit which cannot be directly evaluated against Vizier.
  - How were these 3-4 specific hard to cover points chosen? Were these the entire set of points that were identified to take more than 5 night of simulation?

## Clarity

The mix of background and definitions in Section 2.1 is very hard to understand as someone who is familiar with the semantics but not so much the terminology of RTL. Specifically, the following points are not clear:
- The inclusion of the phi nodes (n4, n8, n12) or generally the parsing of RTL into the graph is not discussed in the text.
- Figure 4, as referenced on Line 107, is in the appendix and not the main body of the paper, which the reference doesn't state.
- Line 99: do the semantics of the execution depend on this nondeterminism? Based on this description, they don't seem to.
- Line 110: "localized" isn't defined here, and I'm not entirely sure what it means. I would have expected a "branch" to be a single edge in the graph, not a triplet of nodes (as it seems to be in Figure 1).
- Line 111: "root" is also not well defined, though it's a bit more possible to infer from context.
- Line 114: After reading this definition of "cover point" I'm still not entirely clear on what it is. Is a "cover point" exactly the same as a branch?
- Line 117: I'm not sure what a "parameter" is -- do these correspond to the parameters in Verilog? Or are these "input"s?
- Figure 1: what is the meaning of the colors? E.g., why are n5-n8 colored blue in most of the figure but green in the "Paths (global)" segment?

Overall, this section would be helped by much more formal definitions, as opposed to the informal English definitions currently used.

Other comments:
- It would be helpful to include some description of the base IPA-GNN architecture in Section 2.3. Also, if the RTL IPA-GNN architecture is claimed as a contribution, it should be described in the main body of the paper.
- Line 212: It would be helpful to mention that Table 7 is in the Appendix. Also, Table 7 does not show 3 real-world designs; it seems to only show "IBEX v1" and "FastML block", and not "IBEX v2".
- For all tables, it would be nice to show some notion of error bars (such as the min and max accuracy across the trials)
- Line 99: there is an extraneous "0" at the end of the line.
- Line 298: This reference to Table 1 should probably be Algorithm 1.
- In general, the paper mixes between content in the main body of the paper and content in the appendix too much, such that it is hard to follow what is being proposed and where to see various tables.


## Significance

I see two primary directions in which this paper could be significant:
- The immediate claims of accurately reducing hardware simulation burden, if borne out, could significantly help hardware design
- The RTL IPA-GNN is an interesting contribution, in that it aims to model the semantics of a language unlike many of the other languages commonly modeled by GNNs (in that it is parallel and nondeterministic). Concretely, RTL IPA-GNN concretely could help future work trying to model RTL, and more abstractly it provides an interesting point of comparison against many other architectures for semantic understanding of code.

## Score

Reject.

While I find the overall motivation and approach compelling, the paper's evaluation and clarity are not sufficient. My highest order concerns are that:
- the paper does not satisfactorily evaluate the claim that "Design2Vec can add practical value to industrial-scale designs"
- the baselines used in Sections 4 and 5 (the MLP and Vizier) are inappropriate for comparing against Design2Vec (in that they neither correspond to reasonable and easily understandable baselines, nor are they representative examples of any prior work on the specific tasks under consideration)
- Section 2 is not clear enough to understand or reproduce the results in the paper.


**Time Spent Reviewing:**

4hr

---

> ### Author Response · Authors · 2021-08-12
> **Response to Higher-Order Concerns**
>
> Thanks for your thorough review. You raise some excellent points. We will respond to your highest-order concerns first, and then we will provide a more detailed response to your other points.
>
> # Practical value
>
> > Section 4: Does not evaluate research question (ii), "whether the performance of Design2Vec can add practical value to industrial-scale designs"
>
> Thanks for the question. Clarifying the end to end value addition will definitely improve the quality of the paper and we will do so. We argue that our results show that there is strong practical value for industrial-scale designs using our approach.
>
> For context, let's take a look at the overall practical flow in industrial verification (Fig 6 and Section 3). Figure 6 shows the state-of-practice flow of RTL design verification. A parameterized test (high level set of parameters as shown in Table 5) is written manually by a verification engineer. Verification engineers do not directly manipulate Boolean instruction sequences (too complex).  Instead, the parameterized test is an input to a complex program called the testbench. The testbench has constraints over some inputs (dictated by legality), while other inputs are generated psuedorandomly. For any given parameterized test, the testbench generates a corresponding legal input stimulus in terms of primary inputs at Boolean level to the design under test (DUT). Each high level parameterized test corresponds to a night of simulation time.
>
> The typical process in industry relies on trial and error. For each cover point, the design verification engineer iterates through the following loop:
>
> 1. Based on the engineer's detailed knowledge of the design, they choose a set of test parameters that are likely to cover the desired cover point. This can require very specialized knowledge about hardware, the testbench, and the DUT.
>
> 1. They wait overnight until the next for the nightly regression in the simulator
>
> 1. Go back to step 1 if the cover point is not covered.
>
> In the industrial designs we consider, this loop might be iterated for thousands of cover points, and hard cover points might require several iterations. So the coverage closure phase as a whole for a single hardware design can often take dozens of person-months.
>
> Design2Vec is intended to expedite this flow in two ways.
>
> First, In coverage prediction (section 4) the model acts as a proxy simulator, allowing the loop to run much more quickly, because step 2 goes from requiring an overnight wait to one call to Design2Vec. Instead, the cycle would become:
>
> 1. Based on the engineer's detailed knowledge of the design, they choose a set of test parameters that are likely to cover the desired cover point. This can require very specialized knowledge about hardware, the testbench, and the DUT.
>
> 1a. The engineer gets the coverage prediction from Design2Vec.
>
> 1b. Repeat steps 1 and 1a until Design2Vec says coverpoint is covered.
>
> 2. The engineer waits until the nightly regression to double-check that the cover points are actually covered (this catches prediction errors from Design2Vec).
>
> 3. If the cover point was not actually covered, go back to step 1.
>
> The 90% accuracy means that step 3 never has to "execute" for 90% of the time. As a general rule in industry, saving one day of time in the hardware design and verification cycle can save multiple $XM in EDA tool licenses, labor, computing resources, opportunity cost and delay in time to market, so shortening this loop is critically important. Chip design and production schedules tend to be delayed by verification across the chip design industry.
>
> Notably, if Design2Vec predicts incorrectly, then this degrades gracefully to the existing process, so there is no additional penalty compared to the current status quo for an incorrect prediction.
>
> Second, the test generation results  in Section 5 show that we can even automate steps 1-3, outperforming state-of-the-art testing approaches.  Design2Vec is able to cover hard to cover points in 17 trials and 2 trials, while the state-of-practice random simulations cannot cover them in nearly 2000 trials (table 4).
>
> # Comparisons to baselines
>
> > the baselines used in Sections 4 and 5 (the MLP and Vizier) are inappropriate for comparing against Design2Vec
>
> ## MLP baseline
>
> We present additional baselines to show that the MLP baseline in Section 4 is useful.
> You are right that the one-hot encoding does not allow the model to generalize to new cover points that are not seen in training. Even so, the MLP can still learn to generalize across test parameters (e.g., some test parameters activate many cover points, some few). Thus, the MLP can perform better than the baseline of naively guessing based on overall statistical frequency.
> We chose the MLP based on the research questions we wanted to evaluate, which included: 1) whether learning techniques can be helpful to learn correlation between test parameters, and 2) whether representation learning of design structure is helpful over no structure. The MLP was meant to provide a baseline for "no representation learning of design structure" (question 2). Also, because it can still learn some correlations between test parameters, the MLP baseline helps to address question 1.
>
> To give more insight into questions 1 and 2, we have added two new baselines for the coverage experiments in Section 4. First, the statistical frequency baseline is the naive baseline of guessing the most common value: Its performance  is the average positive rate over all cover points in the validation data, or one minus the average positive rate, whichever is larger. Unlike the MLP, this baseline does not take into account any correlations between test inputs.
>
> Second, “node seq embedding” is a stronger baseline that enhances the MLP in a way that allows it to generalize to new cover points. Recall that every cover point is defined as a sequence of nodes down a control-flow path from the root of an always block to a particular node, e.g., "n1-n2-n4" in Figure 1. We use the node sequences over all the training cover points to learn in a word2vec model to learn embeddings for each of the control flow nodes, which are concatenated and padded into a cover point embedding. This representation of the cover point is then used in place of the one-hot representation in the MLP.  This way, cover points that have structural proximity in the CDFG graph, e.g., "n1-n2-n4" and "n1-n3-n4", tend to have similar embeddings, so this baseline can generalize to new cover points if there is a nearby coverpoint (in terms of control flow) in training. So this baseline takes some graph structure into account (question 2), but it does not have the full flexibility of GNN-style message propagation.
> We provided below comparisons of all the methods, namely "MLP", "Design2Vec", "Statistical frequency", "Node seq embedding", for each of the three designs.
>
> We observe that Design2Vec is able to learn both structure and test correlation and outperforms each of the three baselines. Also on IBEX V1 and IBEX V2, MLP indeed performs better than the statistical frequency, showing that it can be stronger than random guessing. Node seq embedding is stronger still, but not as strong as Design2Vec.
>
> We will add both of these baselines to the paper --- thanks again for raising this point!
>
> ||IBEX V1|||IBEX V2|||FastML|||
> |-|-|-|-|-|-|-|-|-|-|
> |Train cover points|50%|80%|90%|50%|80%|90%|50%|80%|90%|
> |Design2Vec|0.742|0.773|0.778|0.734|0.780|0.803|0.905|0.906|0.911|
> |Node seq embed|0.597|0.591|0.632|0.585|0.573|0.590|0.879|0.884|0.886|
> |MLP|0.575|0.568|0.568|0.587|0.580|0.582|0.428|0.425|0.347|
> |Statistical Frequency|0.505|0.516|0.508|0.541|0.545| 0.547|0.685|0.686|0.686|
>
> FastML has a regular structure with multiple repeated units, causing the CDFG subgraph to be structurally similar across different cover points. This is probably why Design2Vec and Node seq embedding have a substantially higher accuracy on this design.
>
> ## Comparison to other academic tools
>
> You raise a great point. We would have loved to do so, and we tried, but other tools were unavailable and the designs on which they were evaluated were much smaller than our FastML design. See the common response.
>
> ## Vizier (Section 5)
>
> This is also a significantly stronger baseline than it may seem. Please see details in the common response.
>
>
> # Section 2 is not clear enough to understand or reproduce the results in the paper.
>
> Thanks for the comments and suggestions, we will incorporate them to improve the presentation. The content in section 2 was meant as a high level primer for Verilog and CDFG construction from Verilog for the average ML reader. We will clarify these questions and provide additional references to guide the reader who wants to understand more about Verilog RTL.
>
> To address your concerns about reproducibility, first, we note that CDFG generation techniques exist in literature and are considered fairly standard. This is not a contribution of our paper. The specific procedure that we use to generate CDFGs follows [2,3]. An open source tool is available that builds CDFGs for Verilog RTL using this approach, which is called GoldMine (https://bitbucket.org/debjitp/goldmine/src/v1.2/). CDFG generation can be found here (https://bitbucket.org/debjitp/goldmine/src/v1.2/src/static_analysis.py). A commonly cited reference on constructing CDFGs from Verilog RTL design is [1]. We will add all of these citations to the paper, which together provide enough information to reproduce the inputs to Design2Vec.
>
> [1] C.Wang et al, "Hybrid cegar: combining variable hiding and predicate abstraction," 2007 IEEE/ACM International Conference on Computer-Aided Design, 2007
>
> [2] L. Liu et al, Word level feature discovery to enhance quality of assertion mining, ICCAD 2012
>
> [3] L. Liu et al, "Efficient validation input generation in RTL by hybridized source code analysis," Design, Automation & Test in Europe, 2011

---

> > ### Comment · Reviewer_D7qc · 2021-08-18
> > **Author response**
> >
> > Thanks to the authors for the very detailed response. The response has addressed many of my concerns, and so I'm correspondingly raising my score to a 6.
> >
> > Specifically, the following of my concerns are addressed:
> > - The comparison to the MLP and other baselines is much improved. I appreciate the clarification of the power of the MLP baseline and the inclusion of the statistical baseline, and the node seq embed baseline makes sense as a non- (or at least, less) structural method that can still generate reasonable predictions.
> >
> > The following of my concerns haven't been explicitly addressed, but are promised in the revision (and I believe are likely to be accomplished in the revision):
> > - The clarity and reproducibility of Section 2. Based on the clarity of the author responses to my questions, I am confident that the revision will be significantly more clear.
> >
> > And, the following concerns are still not addressed but I don't think rise to the level of rejecting the paper (due to the good faith effort and failure of the authors to find the appropriate baselines):
> > - The claim that "Design2Vec can add practical value to industrial-scale designs". Though I appreciate the authors' further detailing of the intended use case, this is still not something that is explicitly evaluated in the paper. To address this, if this claim is kept in the paper I would be satisfied with either of the following:
> >   - Explicit case studies or other experimental evaluation demonstrating that for DUTs of interest, Design2Vec results in an improved workflow (ideally compared to other directed testing approaches, though since other directed testing approaches can't be applied to these DUTs another baseline would be fine)
> >   - A comparison against other directed testing approaches showing that the Design2Vec has higher accuracy (though I understand that this isn't possible)
> > - The comparison against Vizier: the authors did not address my concern about the direction of the comparison (in which they use Vizier, a random testing method, as a directed testing method and conclude that it performs worse than Design2Vec, a directed testing method). I would appreciate an additional comparison in the reverse direction, in which Design2Vec is used as a random testing method. For instance, it should be possible to use Design2Vec to target the set of cover points that Vizier discovers to show that Design2Vec can find the same points, or to use Design2Vec with an optimization objective that searches for unseen cover points. I wouldn't be surprised to see Vizier outperform Design2Vec in these benchmarks, but seeing the differences in performance could help to better understand the tradeoffs between the techniques.

---

> ### Author Response · Authors · 2021-08-12
> **Response to Comments about Quality and Clarity**
>
> Thanks again for all the feedback. In this message we respond to your more detailed concerns about quality and clarity. Our other message has responded to your higher-order concens.
>
> # Quality
>
> > rationale behind the choice to include the test inputs at the final MLP rather than earlier in the GNN
>
> Regarding the test inputs, the reasons for including the test inputs at the MLP are to compute a representation of a _model_ in isolation from its inputs, allowing the potential for the Design2Vec representations to be repurposed for new tasks beyond coverage prediction and test generation. We expect such representations are useful for several other tasks such as root cause analysis, debugging and other types of stimulus generation tasks (like generating testbench constraints). Specifically for coverage prediction we agree with your assessment that including the test inputs early in the architecture would be a sensible design decision, and we thank you for that suggestion.
>
> > The RTL IPA-GNN is claimed as a contribution, but a formal description is not in the main body of the paper. There is also no ablation to compare it to the original IPA-GNN.
>
> We will be glad to move the formal description of the RTL IPA-GNN from the appendix to the body of the paper, as per your suggestion. The original IPA-GNN isn't suitable for processing the RTL CDFGs, as it requires only a single-entrance CFG as input to train.
>
> > There are not enough details provided on Algorithm 1 / Figure 3 to reproduce this work.
>
> Thank you for your comments. There was a small mistake in Algorithm 1. In Algorithm 1 there should be an additional step inside the while loop which selects a random cover point from the set ‘uncov’. The objective function is computed for this cover point. By "adjusting the parameter values", we refer to performing gradient ascent (see line 6 in Algorithm 1). We will clarify this in the paper.
>
> > How were these 3-4 specific hard to cover points chosen?
>
> The above definition of hard to cover points pertains to the typical manual state of practice of testing using random stimulus. Due to the higher level of abstraction at which these tests are applied, the manual effort in tuning parameters may not be a good indicator of points that are difficult to reach in the design state space. Eg: a parameter that sets non maskable interrupts may not trigger them at the frequency (another parameter) required to cover a particular interrupt service routine in the design.
>
> We ran a version of the Vizier Smart Regression Plan (SRP) tool that is much more efficient at coverage convergence than random stimulus, After a reasonable number of iterations (200 for IBEX, 243 for FastML blocks), when > 75% cumulative coverage was reached, we selected the remaining cover points (23 out of 900 for IBEX) and declared them as hard to cover (Section 5, para 4 of paper). Given the automatic nature of Vizier SRP's tuning, this approach likely exposes the design states that need specific, infrequent instruction sequences to cover them.
>
> # Clarity
>
> Thanks for all of the great comments about improving the clarity of the paper. Since these are more detailed (which we appreciate very much!), we are replying to them in a separate comment.
>
> These are all good editorial comments. Thanks for pointing them out. We will address these in the paper.
>
> Also, apologies for relegating some of the material to the appendix. We made that call since we wanted to highlight the main points in the body of the paper, and due to space constraints pushed out the details to the appendix. We will do a more comprehensive integration of material so the readability is not affected.
>
> > Figure 4, as referenced on Line 107, is in the appendix and not the main body of the paper, which the reference doesn't state.
>
> We apologize for the oversight. Fig 4 was originally in the main body and moved due to space constraints.
>
> > Line 99: do the semantics of the execution depend on this nondeterminism? Based on this description, they don't seem to.
>
> The semantics at the behavioral level of Verilog RTL (in which we operate) do not depend on nondeterminism. The common practice at this level is to use dynamic validation using simulation kernels, which fix a scheduler in order to obtain an execution trace of the system.
>
> While a formal analysis (like a symbolic model checking) of all possible nondeterministic behaviors is desirable to avoid unexpected execution results at the lower (gate) levels of design, this is not feasible due to scale limitations. Some other approaches have been tried at the gate level. (M. Raffelsieper et al, Formal Analysis of Non-Determinism in Verilog Cell Library Simulation Models)
>
> > Line 111: "root" is also not well defined, though it's a bit more possible to infer from context.
>
> Verilog designs follow a hierarchical modular structure. A root node is an initial statement (after begin keyword) in the top module of the design. A leaf node corresponds to the final executed statement (before the end keyword) in the lowest module in the modular hierarchy. Note that these are non terminating models, implying that there is an implicit loop from the leaf (end) to the root (begin) of every concurrently executing block.
>
> > Line 114: After reading this definition of "cover point" I'm still not entirely clear on what it is. Is a "cover point" exactly the same as a branch?
>
> Every node in the CDFG is a statement in the RTL. A branch is a sequence of nodes (it could also be defined as a sequence of edges). We distinguish between a localized branch and a global path, since coverage in Verilog simulators is measured in terms of branches. Path coverage is not measurable/available in state of practice tools. A localized branch (or branch) is a sequence of nodes starting from a control node (if, case, casex statements). A global path is a sequence of nodes starting from a root node of the graph and ending with a leaf node of the graph. As shown in Fig 1, n1-n2-n4 is a local branch and n1-n2-n4-n9-n10-n12 is a global path.
>
> Cover points are defined for different code constructs based on coverage metrics defined for RTL verification. Branch cover points correspond to branch coverage. Line cover points correspond to line coverage, finite state machine cover points correspond to FSM coverage, state transition cover points to state transition coverage and assertion cover points to assertion coverage. In this work, we consider only branch cover points, hence the definition of a cover point is the same as a branch. In future, we plan to extend this to other code coverage metrics.
>
> > Line 117: I'm not sure what a "parameter" is -- do these correspond to the parameters in Verilog? Or are these "input"s?
>
> These are input parameters for a test. A test is a set of parameters. Each parameter can be of Boolean, integer or categorical.
> From the example in Table 5, the test generated by Design2Vec has the parameters +instr_cnt, +illegal_instr_ratio, +enable_unaligned_load_store… .
> instr_cnt (integer) is the instruction count or the number of instructions that should be part of the given test.
> Illegal_instr_ratio (integer) is the ratio of illegal instructions to legal instructions that should be part of the given test
> enable_unaligned_load_store (Boolean) is a flag that enables unaligned memory accesses or disables them for a given test
>
> > Figure 1: what is the meaning of the colors? E.g., why are n5-n8 colored blue in most of the figure but green in the "Paths (global)" segment?
>
> The colors of nodes in Figure 1, Figure 4 and Figure 5 are meant to denote different concurrently executing (always) blocks. In Figure 1, n1, n2, n3 and n4 in yellow correspond to the always blocks in yellow. n5, n6, n7, n8 correspond to the always block in blue. n9, n10, n11, n12 correspond to the always block in green. Each always block will be evaluated in every cycle.
>
> Colors of arrows are meant to denote specific execution paths in Figures 4 and 5. The specific execution path taken depends on the inputs to the module in that cycle, or from previous cycles. For example, in Figure 4, the green always block executes with a and b values from previous cycle t. The other two blocks execute based on inputs in current cycle ẗ+1. (section 2.1 para 4).
>
> In figure 5, the colors of the arrows again denote the execution path taken for the given input stimulus.
> (Note that we do not generate direct Boolean stimulus to the design in this problem. We generate a parameterized test. The parameterized test is an input to a testbench program, whose output is the input stimulus that is applied directly to the design (as in Fig 6).  )
>
> > Overall, this section would be helped by much more formal definitions, as opposed to the informal English definitions currently used.
>
> Thanks for the comments. As space permits, we will add more detailed and formal definitions in section 2.1 of the paper, along with corresponding references to standard Verilog to CDFG conversion techniques in literature.

---

### Author Response · Authors · 2021-08-12
**Common response: Comparison with other test generation tools**

Thanks to all of the reviewers. There were a few common points across reviews that we wanted to reply to centrally.

## Test generation: Why compare to Vizier and not other academic work?

  There are two kinds of automatic RTL test generation techniques in academic literature - random test generation based and directed test generation based. Random test generation refers to undirected approaches to test the design maximally (similar to fuzzing in software). Directed test generation refers to targeting specific code or functional cover points in the design and generating tests for them. Both these approaches are practically useful - random test generation is applied to achieve best effort cumulative coverage. Directed testing is meant for targeting the hard to cover points that the random testing cannot cover. Since these two approaches optimize different metrics, cumulative coverage is the only way to compare them even though it might not be as favorable to directed test generation approaches.

**Comparison with Vizier:** Our approach for testing with Design2Vec is a directed testing approach, since we provide specific cover point(s) and Design2Vec generates a test for them. [Vizier Smart Regression Planner (SRP)](https://capra.cs.cornell.edu/latte21/paper/31.pdf) is a dedicated tool for RTL test generation that adapts the original Bayesian optimization based tool (Vizier) for this problem. Vizier SRP is a random testing based approach that uses Bayesian optimization with cumulative coverage as an objective. As indicated in that paper, it is a state-of-the-art tool that is being used at an industrial scale. Hence, Vizier SRP presents a much stronger baseline than an academic tool that has not been shown to generate tests for practical designs. Despite the disadvantage to Design2Vec in comparison with a random testing based tool that optimizes for cumulative coverage, we present Vizier SRP, as a tough baseline comparison.

**Comparison with other academic tools:** We tried to compare Design2Vec with the three recent and most prominent academic papers mentioned in the references - DirectFuzz, RFuzz and Concolic testing for RTL. Of them, RFuzz is random testing, and the other two are directed testing based tools. Please note that none of the academic RTL test generation methods above have shown the scale that we need (with FastML) for a baseline comparison. Their designs are smaller by two orders of magnitude than FastML block (10K gates instead of 1M gates). We were trying to compare the smaller IBEX benchmark with respect to quality (coverage) instead of scale. Since 2/3 of these tools were not available publicly for comparison, we also sent them our IBEX benchmark design RTL  and results on branch coverage using the VCS simulator that they could run on their end and report results on. Unfortunately, the input format and the metrics used by them are not standard, and are as such hard to compare with our results. The following are paraphrased versions of author responses:

- Concolic testing for RTL tool and infrastructure has not been placed in the public domain since it is being funded by some classified agencies.
- DirectFuzz (based on RFuzz) is not available for public use. It has been run only on designs that were presented in the paper. The IBEX RTL could not be parsed into the intermediate format since it has many new constructs. The authors said they could not generate results for our benchmarks using their tool at this time.
- RFuzz is available publicly. However, it does not work directly on Verilog RTL or the commercial standard VCS simulator. It needs Verilog RTL to be converted into an intermediate format. The IBEX RTL could not be parsed into the intermediate format since it has many new constructs. The author of RFuzz suggested that we mention the tool is not available for comparison at this point. They were working on adding new constructs into their parser. Also, the reporting of coverage in this tool is not in terms of branch coverage, but a different metric not used by standard industrial tools. Comparison with this tool required us to hack into their parser and add constructs that could parse Ibex RTL. We attempted this as well, but without much success.

**Commercial tools:** To our knowledge, there are no commercial EDA tools that provide RTL test generation as a mainstream feature that we can use and compare our tool against.

---

### Decision · Program_Chairs · 2021-09-27

**Decision:**

Accept (Poster)

**Comment:**

The idea of predictive coverage/testing for RTL designs using modern representation learning methods is interesting and potentially useful. Most of the reviewer discussion surrounded the best ways to evaluate such a method; eventually reviewers agreed that the authors did as well as could be expected given the state of open tools for verification -- this broader community issue should be mentioned in the revision. Some reviewers also requested clarifications and details that should be incorporated in the revision. Overall a good step in this application space.